# Dynamin Inhibitors Prevent the Establishment of the Cytomegalovirus Assembly Compartment in the Early Phase of Infection

**DOI:** 10.3390/life11090876

**Published:** 2021-08-25

**Authors:** Igor Štimac, Natalia Jug Vučko, Gordana Blagojević Zagorac, Marina Marcelić, Hana Mahmutefendić Lučin, Pero Lučin

**Affiliations:** 1Department of Physiology and Immunology, Faculty of Medicine, University of Rijeka, 51000 Rijeka, Croatia; igor.stimac@uniri.hr (I.Š.); natalia.jug@uniri.hr (N.J.V.); gordana.blagojevic@uniri.hr (G.B.Z.); mmarcelic@uniri.hr (M.M.); pero@uniri.hr (P.L.); 2Nursing Department, University North, University Center Varaždin, Jurja Križanića 31b, 42000 Varaždin, Croatia

**Keywords:** cytomegalovirus, assembly compartment, dynamin, early endosomes, endosomal recycling compartment, Golgi apparatus

## Abstract

Cytomegalovirus (CMV) infection initiates massive rearrangement of cytoplasmic organelles to generate assembly compartment (AC). The earliest events, the establishment of the preAC, are initiated in the early phase as an extensive reorganization of early endosomes (EEs), endosomal recycling compartment (ERC), trans-Golgi network (TGN), and the Golgi. Here, we demonstrate that dynamin inhibitors (Dynasore, Dyngo-4a, MiTMAB, and Dynole-34-2) block the establishment of the preAC in murine CMV (MCMV) infected cells. In this study, we extensively analyzed the effect of Dynasore on the Golgi reorganization sequence into the outer preAC. We also monitored the development of the inner preAC using a set of markers that define EEs (Rab5, Vps34, EEA1, and Hrs), the EE-ERC interface (Rab10), the ERC (Rab11, Arf6), three layers of the Golgi (GRASP65, GM130, Golgin97), and late endosomes (Lamp1). Dynasore inhibited the pericentriolar accumulation of all markers that display EE-ERC-TGN interface in the inner preAC and prevented Golgi unlinking and dislocation to the outer preAC. Furthermore, in pulse-chase experiments, we demonstrated that the presence of dynasore only during the early phase of MCMV infection (4–14 hpi) is sufficient to prevent not only AC formation but also the synthesis of late-phase proteins and virion production. Therefore, our results indicate that dynamin-2 acts as a part of the machinery required for AC generation and rearrangement of EE/ERC/Golgi membranes in the early phase of CMV infection.

## 1. Introduction

Cytomegaloviruses (CMV) are large DNA viruses that belong to a family of β-herpesviruses and cause life-long asymptomatic infections in around 70% of the population. However, after congenital infections and infections in immunocompromised patients, the virus severely damages the infected organism, often with the fatal outcome. A vaccine or adequate antiviral therapy against CMV is still not established [1,2]. Therefore, understanding the mechanism(s) that viruses undertake in infected cells to produce their progeny is of outstanding importance.

In the first phase of β-herpesvirus replication, nucleocapsids are assembled in the cell nucleus and exit into the cytoplasm by the well-defined process of envelopment and de-envelopment across the inner and outer nuclear membranes, respectively. On the other hand, although extensively investigated [3,4,5], mechanisms of the final envelopment and the virion egress from the cell are still insufficiently understood. It is well established that massive reorganization of membranous organelles is essential for developing the sites required for final envelopment and egress of nascent virions. These rearrangements include early endosomes (EE), endosomal recycling compartment (ERC), the trans-Golgi network (TGN), and the Golgi system, both in HCMV [6,7,8] and MCMV [9,10] infected cells. Together with viral tegument and envelope proteins, this structure is known as the assembly compartment (AC) [6,8,11]. The process of membranous organelle remodeling is initiated very early in the infection [9,10,12]. It involves altering many host-cell factors that regulate membranous organelle trafficking (i.e., small GTPase from the Rab and Arf subfamilies and their regulators and effectors) [12]. The reorganized membranous organelle structure in the early phase of infection, before viral DNA replication and expression of viral structural proteins is considered the preAC [10,13].

The (pre)AC includes the pericentriolar accumulation of vesicular, vacuolar, and tubular membranous elements of EE, ERC, and TGN origin surrounded by dislocated and reorganized Golgi stacks [4,6,10,14]. These membranous structures accommodate many membrane-associated viral proteins expressed in the early phase of infection, such as m06 protein [15]. It is still unresolved if EE-ERC-TGN-derived membranous vesicles and tubules represent completely rearranged original cell compartments with a new mixed composition (i.e., Rab/Arf GTPases and their effectors), or cellular compartments are just relocalized and maintain their initial integrity. To resolve this dilemma, it is essential: (i) to fully understand the spatial-temporal relationships in the membranous system of a healthy cell, including identification and characterization of all molecular mechanisms that drive membrane flow between compartments; (ii) to perform a detailed analysis of AC membranes in infected cells; and (iii) to integrate the knowledge acquired in healthy and infected cells. Considering that numerous communication processes between cellular membranes, even in non-infected cells, are still poorly understood [16,17], it is evident that the clarification of AC functioning is a challenging task.

Many processes in membranous organelle trafficking are regulated by dynamin, a large cytoplasmic protein that belongs to the dynamin-related superfamily (DRS) of large GTPases [18]. Three dynamin genes are found in mammalian cells (DNM1, DNM2, and DNM3). While dynamin-2 is expressed ubiquitously, dynamin-1 is expressed in the nervous system and brain, and dynamin-3 in the brain, testes, and megakaryocytes [19,20]. As a scission protein, dynamin binds around the neck of detaching vesicle, constricts, and finally cuts it off. Besides its function in endocytosis, where it has a role in the fission of endocytic vesicles from the plasma membrane [21,22,23], dynamin is essential for vesicle scission from Golgi [24,25,26], exocytosis [27], cell motility, and cytokinesis [28,29,30,31,32]. Its role in endosomal recycling is still intriguing, and the study of Mesaki et al. claims that dynamin is essential for detaching recycling tubules from EEs [33].

Many approaches have been used to explore the dynamin role. The overexpression of dominant-negative Dyn-2 mutant K44A [21] is used the most frequently, but other Dyn-2 mutants that target their GTPase activity—such as S45N, T65F, T65A, K694A, S61D, and T141A—are also applied [34,35]. The knockdown of dynamin with small interference RNA (siRNA) [31,36] and knockout of one, two, or all three dynamin genes [37,38] also provided valuable insight into the dynamin function. Also, the treatment with cell-permeable dynamin inhibitors is a widely accepted approach [39,40], mainly due to their important advantage of rapid activity and reversibility, which enable scientists to perform short-term pulse-chase experiments. The most widely used are dynasore, a non-competitive inhibitor of the dynamin GTPase activity [41], and Dyngo compounds (such as Dyngo-4a), dynasore analogs with improved potency and reduced off-target effects [42,43]. In addition, Dynoles (such as dynole 34-2), which also inhibit the dynamin GTPase activity [44], and MiTMAB (myristyl trimethyl ammonium bromides), which prevent dynamin mobilization to the cell membranes [45], are often used to study dynamin function.

Although many studies addressed the role of dynamin in the endocytic entry of various viruses [39], the dynamin role in the post-entry processes of viral infection is poorly characterized. Several studies demonstrated the contribution of dynamin in the trafficking of viral proteins, such as HBsAg budding from infected hepatocytes [46], vesicular stomatitis virus glycoprotein (VSV-G) release from the TGN [36], and endocytosis of HSV-1 glycoproteins from the plasma membrane into HSV-1 assembly compartments for virion production [47]. Finally, the impact of dynamin on cytomegalovirus maturation and egress has been confirmed for HCMV [48] and MCMV [38].

The present study is focused on the role of dynamin in the first phase of the AC establishment during the early phase of MCMV infection, which occurs in the period of 6 to 14 h post-infection (hpi) in fibroblasts and can be defined as the pre-AC. We explored the effects of the dynamin inhibitor dynasore on the pericentriolar reorganization of EEs, the EE-ERC intermediate compartment, and the ERC into the membranous organelle agglomerate representing the inner preAC (i-preAC) and the dislocation of cis/medial/trans-Golgi cisternae into the outer ring representing the outer preAC (o-preAC). Our study demonstrates that short treatment with dynasore in the early phase of infection abolishes the establishment of the pre-AC, prevents the synthesis of late-phase MCMV proteins, and inhibits virus production. Therefore, our results strongly indicate that dynamin plays an essential role in the early phase of CMV infection, in CMV-induced membranous organelle reorganization that establishes the AC. These results contribute to a better understanding of CMV biology and options of CMV therapies, as dynamin inhibitors are already investigated to treat different diseases [49,50].

## 2. Materials and Methods

### 2.1. Cell Lines, Viruses, and Infection Conditions

All experiments were performed on Balb3T3 fibroblasts (American Type Culture Collection, clone A31, ATCC CCL-163, Manassas, VA, USA). Primary murine embryonic fibroblasts (MEFs), generated from 17 days embryos of BALB/c mice, served for virus production and titration experiments. Cells were cultured in DMEM supplemented with 10% (5% for MEF) of fetal bovine serum (FBS), 2 mM L-glutamine, 100 mg/mL of streptomycin, and 100 U/mL penicillin (all reagents from Gibco/Invitrogen, Grand Island, NY, USA) at 37 °C with 5% CO_2_. When used for the propagation, cells were cultured in 10 cm Petri dishes and split when 80–90% confluent. Production of MCMV stocks and infection of cells have followed the standard procedure as previously described [51]. Cells were infected with 1 plaque-forming unit (PFU)/cell with a multiplicity of infection (MOI) of 10 after enhancement of infectivity by centrifugation [51].

To avoid the interference of MCMV FcR with tested antibodies and false-positive results, the recombinant virus Δm138-MCMV (ΔMC95.15), with the deletion of the fcr1 (m138) gene [52] was used in all infections. To quantify virus growth by the plaque assay, we used wt MCMV (strain Smith, ATCC VR-194).

### 2.2. Antibodies and Reagents

The following antibodies were used for labeling of markers of endosomal compartments: rabbit monoclonal IgG anti Rab10 (Cat.No. 8127; Cell Signaling Inc, Danvers, MA, USA), mouse monoclonal IgG_1_ anti GM130 (Cat.No. 610823, BD Biosciences, Franklin Lakes, NJ, USA), rabbit monoclonal anti Rab5a (Cat.No. 3547, Cell Signaling Inc, Danvers, MA, USA), rabbit monoclonal IgG anti Vps35 (Cat.No. 4263; Cell Signaling Inc, Danvers, MA, USA), chicken polyclonal anti EEA1 (Cat.No. 40-5700, Invitrogen, Thermo Fisher Scientific, Waltham, MA, USA), rabbit monoclonal IgG anti Hrs (Cat.No. 15087; Cell Signaling Inc, Danvers, MA, USA), rabbit monoclonal IgG anti Rab11 (Cat.No. 5589; Cell Signaling Inc, Danvers, MA, USA), rabbit monoclonal IgG anti Arf6 (Cat.No. 5740; Cell Signaling Inc, Danvers, MA, USA); rabbit polyclonal anti GRASP65 (Cat.No. PA3-910, Thermo Fisher Scientific, Waltham, MA, USA), rabbit polyclonal Golgin 97 (Cat.No. ab84340, Abcam, Cambridge, UK), rat monoclonal anti Lamp1 IgG_2a_ (Cat.No. 553792, BD Biosciences, Franklin Lakes, NJ, USA). Antibodies for MCMV-encoded proteins are produced by the University of Rijeka Center for Proteomics. All anti-MCMV antibodies have been verified at the University of Rijeka Center of Proteomics (https://products.capri.com.hr/shop/?swoof=1&pa_reactivity=murine-cytomegalovirus; Accessed on 24 Aug 2021), and in our laboratory. They included: mouse monoclonal IgG_1_ anti m123/IE1 (CROMA101), mouse monoclonal IgG_2a_ anti m123/IE1 (clone IE1.01), mouse monoclonal IgG_1_ M55/gB (clone M55.01 for Western blot), mouse monoclonal IgG2a M55/gB (clone M55.02 for immunofluorescence), mouse monoclonal IgG_1_ anti M25 (M25C.01), mouse monoclonal anti M57 (clone M57.02), mouse monoclonal IgG anti M74 (clone 74.01). Secondary antibodies: AF^488^-, AF^594^-, and AF^555^-conjugated secondary antibody reagents to mouse IgG_2a_, mouse IgG_2b_, mouse IgG_1_, rat IgG, rabbit IgG, and chicken IgG were from Molecular Probes (Leiden, The Netherlands), and AF^647^-conjugated IgG_1_ and IgG_2a_ were from Jacksons Laboratory (Bar Harbor, ME, USA).

Inhibitors of endocytic transport were as follows; Dynasore (Cat.No. D7693, Sigma Aldrich, St. Louis, MO, USA), Dyngo-4a (Cat.No. ab120689, Abcam, Cambridge, UK), MiTMAB (Cat.No. ab120466, Abcam, Cambridge, UK), Dynole 34-2 (Cat.No. ab120463, Abcam, Cambridge, MA, USA), Pitstop (Cat.No. ab120687, Abcam, Cambridge, UK), methyl-β-cyclodextrin (Cat.No. C4555, Sigma-Aldrich Chemie GmbH, Taufkirchen, Germany). DAPI (4′,6-diamidino-2-phenylindole, dihydrochloride) was from Thermo Fisher Scientific, Waltham, MA, USA; (Cat.No. D1306). Other chemicals were from Sigma-Aldrich Chemie GmbH (Taufkirchen, Germany).

### 2.3. Immunofluorescence, Confocal Microscopy, and Image Analysis

Cells were grown on coverslips in 24-well plates and were proceeded for immunofluorescence when 60–70% confluent. After fixation with 4% formaldehyde (20 min at r.t.), and permeabilization with 1% Tween 20 (20 min at 37 °C), cells were incubated with primary antibodies (60 min at r.t.), washed three times with PBS, and incubated with appropriate fluorochrome-conjugated secondary antibodies (60 min at r.t.). After washing in PBS, cells were embedded in Mowiol (Fluka Chemicals, Selzee, Germany)-DABCO (Sigma Chemical Co, Steinheim, Germany) in PBS containing 50% glycerol and analyzed by confocal microscopy.

Imaging was performed on Leica DMI8 inverted confocal microscope (confocal part: TCS SP8; Leica Microsystems GmbH, Wetzlar, Germany) equipped with a UV (diode 405), Ar 488, DPSS 561, and He/Ne 633 lasers, and 4 detectors (2 × PMT and 2 × HyD). HC PLAPO CS2 (63×/1.40 oil) objective was used. The images (515 × 515 pixels) were acquired by LAS (Leica Application Suite) X software (Leica Microsystems GmbH, Wetzlar, Germany), in a sequential mode with z-series of 0.5 μm (offset 0–1.5%), and zoom factors of 0.75× (pixel size 481.47 × 481.47 nm), 1.5× (pixel size 240.74 × 240.74 nm), 3× (pixel size 120.37 × 120.37 nm), and 6× (pixel size 60.18 × 60.18 nm). Some images in Section 3.6 were acquired with Olympus Fluoview FV300 confocal microscope (Olympus Optical Co., Tokyo, Japan) equipped with Ar 488, He/Ne 543, and He/Ne 633 lasers and Fluoview software, version 4.3 FV 300 (Olympus Optical Co., Tokyo, Japan), PLAPO60xO objective, appropriate barrier filters, and PMT detectors. Images were taken under the controlled parameter settings and exported in TIFF formats.

The ratio of infected cells with AC (AC was defined as any fluorescent signal that was concentrated in the pericentriolar area within the angle of α ≤ 90 explained in Section 3.1 and Section 3.4, was calculated by direct image counting at least 10 fields of view per one immunofluorescence sample (200–400 cells) on epifluorescent Olympus BX51 microscope equipped with DP71CCD camera (Olympus, Tokyo, Japan) with UPlanFL N 40×/0.75 objective.

Images were analyzed using ImageJ 1.53c software and available plugins.

### 2.4. Flow Cytometry and Quantification of Total M55/gB Protein

Cells were detached by short trypsin/EDTA treatment, washed in PBS containing 10 mM EDTA, HEPES pH = 7.2, 0.1% NaN_3_, and 2% FCS (PBS-A), fixed and permeabilized with BD Cytofix/cytoperm (Cat.No. 51-2090KZ, BD Biosciences, Franklin Lakes, NJ, USA) (15 min, r.t.), washed three times in 0.1% saponin, and labeled with anti M55/gB primary antibody (clone M55.02), and anti-mouse IgG2a Alexa 488 secondary antibody (all antibodies dissolved in 0.1% saponin). Following washing in PBS-A, cells were analyzed using FACSCalibur flow cytometer (Becton Dickinson & Co, San Jose, CA, USA), where 5000 cells were acquired per sample. The fluorescence signal was expressed as ΔMFI − mean fluorescence intensity (MFI) of infected cells after subtracting fluorescence signal calculated from non-specific binding of M55.02 antibody to uninfected cells treated in the same way.

### 2.5. Western Blot

Whole cellular lysates were prepared in RIPA lysis buffer supplemented with protease inhibitors (Cat.No. 11697498001, Roche Diagnostics GmbH, Unterhaching, Germany) and proceeded for SDS-PAGE electrophoresis (Bio-Rad PowerPac Universal, Hercules, CA, USA), and blotting (Bio-Rad Trans-Blot Turbo Transfer System, Hercules, CA, USA) onto polyvinylidene difluoride (PVDF-P) WB membrane (Millipore) at 60 to 70 V for 1 h. Following blocking in 1% blocking reagent (Roche Diagnostics GmbH, Mannheim, Germany) for 1 h, and incubation with appropriate primary antibodies (1 h to overnight at 4 °C), membranes were washed three times with T-TBS buffer (TBS with 0.05% Tween 20; pH = 7.5) and incubated with peroxidase-conjugated secondary antibody diluted in TBS buffer containing 0.5% blocking reagent (45–60 min, r.t.). After final washing in TBS-T buffer, the chemiluminescent signal was achieved after incubation with SignalFire (TM) Plus ECL Reagent or SignalFire (TM) Elite ECL Reagent (Cell Signaling, Cat.No. 12630S, 12757P) for 1 min, and detection by Transilluminator Alliance 4.7 (Uvitec Ltd., Cambridge, UK). Mouse anti-β-actin (Millipore, Billerica, MA, USA) was used as a control of protein loading.

Western blot signals were quantitatively analyzed by Image J 1.53 software and normalized to actin signal that was used as a loading control. We first calculated the normalization factor for every lane according to the formula: Lane normalization factor = Observed actin signal for every lane/Highest observed signal of actin for the blot. After that, normalized experimental signals were calculated as the ratio of the observed experimental signal/lane normalization factor. The relative level of 1 represents the maximal signal within the Western blots at the same membrane.

### 2.6. Quantification of Virus Growth by the Plaque Assay

Balb3T3 fibroblasts were cultured in 24-well plates in duplicates (approx. 9 × 10^4^ cells/well) and infected with Δm138-MCMV at MOI of 10. Cell culture medium with unbound viruses was removed at 4 hpi and replaced with the medium containing solvent (0.1% DMSO in 10% DMEM) or inhibitor (80 µM Dynasore in 10% DMEM). In the half of dynasore-treated, the cell culture medium was replaced with the standard medium (0.1% DMSO in 10% DMEM) at 14 hpi. After 48 and 72 hpi, supernatants were harvested, and the number of released virus particles was determined by the standard three-replicate plaque assay on MEFs in 48-well plates [51].

### 2.7. Cell Viability

Balb3T3 fibroblasts infected with Δm138-MCMV were cultured in 24-well plates and treated with inhibitors as presented in Section 3.3. After determined time points (14 hpi and 40 hpi), cell viability was analyzed with trypan blue exclusion assay as described [53].

### 2.8. Data Presentation and Statistical Analysis

The data are presented as mean ± standard deviation of the mean. To determine significynce of difference, we used Student’s *t*-test. Differences are indicated on diagrams as significant when *p* values were <0.05 (* *p* < 0.05; ** *p* < 0.01; *** *p* < 0.001).

## 3. Results

### 3.1. The Time Course of the preAC Establishment

We have previously shown that reorganization of host-cell compartments is initiated in the early phase of MCMV infection and results in the development of preAC [10]. This primordial structure finally develops into mature AC, characterized by pericentriolar accumulation of EE-, ERC-, and TGN-derived membranous structures in the inner area (inner AC; iAC) surrounded by reorganized Golgi stacks into the outer ring-like formation (outerAC; oAC).

To find an optimal experimental setup for analysis of the preAC, we first monitored the course of preAC development in the early phase of infection by immunofluorescence. Given that Rab10, a marker of EE/ERC intermediates [54,55], does not label significant organelles in uninfected Balb 3T3 cells and highly recruits to membranes of the innerAC [10], it was selected as the earliest marker that distinguishes i-preAC in immunofluorescence studies. The development of the o-preAC, represented as dislocation of Golgi cisternae into the outer ring [10,56], was monitored by visualization of GM130 at cis-Golgi cisternae.

As expected, Rab10 was not recruited to membranous organelles of substantial size in uninfected cells (Figure 1A). Although the Rab10 signal slightly increased in 4 h infected cells, it remained diffuse and without signs of a substantial concentration (Figure 1A). However, in 6 h infected cells, Rab10 accumulated in the pericentriolar area (Figure 1A) of 33.3 ± 2.9% of IE1^+^ cells (Figure 1B). The Rab10 accumulation increased in size with the progression of infection, and at 14 hpi, 72.1 ± 3.6% of IE1^+^ cells expressed Rab10 in the i-preAC (Figure 1A,B). These results demonstrate that the i-preAC cannot be accurately visualized before 6 hpi, which is present in most cells at the end of the early phase of MCMV infection.

We next analyzed changes in the Golgi pattern during the early phase of infection. For precise Golgi analysis, we adapted patterns described by Wollrab et al. [57], who recognized extended, compacted, and fragmented Golgi patterns, and Rebmann et al. [56], who identified intact, fragmented, and AC^+^ Golgi in HCMV infected cells. Therefore, as shown in Figure 1C (right panel), we divided Golgi appearance into four patterns: (A) Extended, α > 90°, (B) Compacted, α < 90°, and (C) Ringlike (typical for AC), α < 90°. The extended A form was further divided into Aa—ordered (intact) and Ab—disordered. Considering that the Golgi fragmentation may occur in all patterns, we did not analyze it as a separate form. In uninfected cells, the extended and ordered pattern (Aa) was present in the majority (67.8 ± 4.2%) of cells and represented standard, cisternal Golgi organization. The extended and disordered (Ab) pattern was identified in 27.4 ± 1.4% cells and compacted pattern B in 4.4 ± 2.3% cells (Figure 1A,C). These patterns are likely associated with the cell entry into the G2-M phase of the cycle when Golgi normally fragments [58,59,60].

As presented in Figure 1A,C, at 4 h of MCMV infection, the pattern Aa (extended and ordered/intact Golgi) was present in 21.0 ± 1.7% cells, most of the cells (51.0 ± 6.5%) displayed the extended but disordered Golgi (pattern Ab) (Figure 1A, arrowhead), and a substantial fraction of cells already displayed compacted Golgi (pattern B, Figure 1A, arrow). The C pattern (ring-like Golgi, Figure 1A, asterisk) was present only in 5.5 ± 3.6% of cells. However, in most the 6 h infected cells, the Golgi was compacted and dislocated from the juxtanuclear area. The 42.8 ± 10.6% cells displayed pattern B, and 23.9 ± 6.3% of cells the pattern C (AC). At 14 hpi, 74.4 ± 7.5% of cells showed the pattern C (Figure 1A,C).

This analysis demonstrates that the outer preAC is developed through the sequence of the Golgi reorganization, which involves unlinking, compacting, and dislocation of the Golgi cisternae. The initiation of this sequence can be identified already at 4 hpi as a disordered Golgi reflecting unlinking of the Golgi stacks. The dislocated Golgi pattern representing outer preAC begins to appear at 5–6 hpi and remains in this form throughout the early phase of infection (Figure 1) as the fully established AC [10]. The appearance of dislocated Golgi (pattern C) coincided with the appearance of Rab10 concentration in the juxtanuclear area (Figure 1A). Thus, visualization of the dislocated Golgi by staining against GM130 protein and concentrated juxtanuclear Rab10 is one of the earliest indications for developing the preAC in MCMV infected cells.

### 3.2. Dynasore Inhibits Establishment of the preAC

To explore the role of dynamin in the early phase of preAC establishment, we needed a tool that can efficiently shut off the dynamin function at the time of initial membranous organelle reorganization (i.e., at 4 hpi) and would not interfere with earlier events in the course of MCMV infection (i.e., establishment of infection, expression of MCMV early-gene function). We used dynasore, a well-established and frequently used small molecule that noncompetitively and reversibly inhibits GTPase activity of the dynamin within 2 min [33,41,48].

When added at 4 hpi to MCMV infected cells, dynasore treatment prevented the pericentriolar accumulation of Rab10 (Figure 2A). Almost none of the infected cells displayed accumulation of Rab10 at 6 hpi (Figure 2, Table 1), two hours after dynasore treatment. At 14 hpi, the Rab10 accumulation can be identified in 5.4 ± 2.2%, whereas at 48 hpi, it can be detected in 10.8 ± 5.8% of infected cells (Figure 2B, Table 1). These data indicate that dynasore efficiently blocks membranous organelle reorganization that forms the inner preAC in the early and the innerAC in the late stages of infection.

After visualization of Golgi membranes with GM130 (Figure 2A), we observed that dynasore treatment also interfered with the process of Golgi reorganization during the early phase of infection. The effect was present throughout the whole kinetics of infection (Figure 2A,B, right panel, and Table 1). Dynasore treatment prevented the establishment of the ring-like (pattern C) form of the Golgi (*p* < 0.05 for 6 hpi, and *p* < 0.001 for 14 and 48 hpi), which represents the o-preAC and the oAC in CMV infected cells. Otherwise, the Ab Golgi pattern (extended but disordered) dominated (55–70% of cells), and the pattern B was found in 20–30% of dynasore-treated MCMV-infected cells (Figure 2A,B, Table 1). This result suggests that dynasore treatment prevents Golgi displacement into AC but also affects the Golgi stability.

Thus, we further examined the effect of the dynasore in uninfected cells (Appendix A). The extended but disordered Golgi pattern (Ab) dominated, which likely developed due to the dynamin effect on the Golgi homeostasis [24,36]. Interestingly, the general effect of dynasore on the Golgi unlinking and fragmentation was similar to that observed in dynasore-treated infected cells (compare Figure 2 and Appendix A). These data indicate that Golgi unlinking and fragmentation, which take place early in MCMV infection, likely represents overture into succeeding Golgi displacement but are not sufficient per se for the final dislocation of the Golgi into the ring-like form. Namely, proceeding into the C pattern could be prevented by dynasore, indicating a dynamin-dependent process in the o-preAC development.

Finally, we tested if the dynasore treatment from the beginning of the infection would have changed Golgi reorganization and preAC formation compared to 4 hpi. However, the ratios in Golgi patterns were similar to that observed after the addition of dynasore at 4 hpi (Appendix A). We only observed less fragmentation and more vacuolization of the Golgi, likely due to inhibition of dynamin that plays a role in detaching vesicles from TGN [24,36].

In summary, this data indicate that dynamin plays an essential role in preAC and AC formation, characterized by inhibition of the pericentriolar accumulation of Rab10 and the Golgi displacement.

### 3.3. Other Dynamin Inhibitors, but Not Clathrin Inhibitor and Cholesterol Depletion, Prevent the Establishment of the preAC

To confirm that dynamin is essential for the preAC establishment and AC formation, we treated cells with three additional rapidly-acting dynamin inhibitors: Dyngo-4a, Dynole 34-2, and MiTMAB. Dyngo-4a is an improved dynasore analog characterized by less toxicity and less unspecific reactions [42,43]. Dynole 34-2, as dynasore and Dyngo-4a, inhibits dynamin activity after recruitment at the cell membranes. It binds the G domain of dynamin and, in a non-competitive way, interferes with its GTP cycle [44]. MiTMABs (myristyl trimethyl ammonium bromides) prevent dynamin recruitment via its PH domains to the cell membranes [45].

To establish the optimal experimental conditions regarding effectiveness and cell viability, we tested different concentrations of the inhibitors in line with data from the literature. The establishment of the preAC was monitored by Rab10 accumulation in the pericentriolar area, and the development of the mature AC was monitored by simultaneous Rab10 accumulation in the iAC and late MCMV glycoprotein B (M55) [61] in the oAC [10]. The effects of selected concentrations of inhibitors are presented in Table 2.

As presented in Table 2 and Figure 3, dynasore and its analog Dyngo-4a efficiently prevented the Rab10 accumulation in infected cells at 16 and 40 hpi. MitMAB and dynole 34-2 inhibited Rab10 accumulation at 16 hpi, but less than dynasore and Dyngo 4a, whereas at 40 hpi, the inhibitory effect was closer to the other two dynamin inhibitors. These data suggest that dynamin activity is essential for Rab10 accumulation in the inner preAC and AC.

To explore their effects on the outer AC, we simultaneously stained infected cells against the MCMV glycoprotein B (M55) at 40 hpi (Table 2). As recently reported, M55 was expressed as a perinuclear ring in 56.6 ± 8.6% of infected cells (Figure 3A), in a similar pattern as the Golgi markers [10]. Importantly, all dynamin inhibitors almost completely prevented visualization of M55 protein in the outer AC, suggesting their additional effects on the expression of MCMV proteins. These data indicate that dynamin inhibition has multiple impacts in the AC biogenesis, in addition to the inhibitory effects on the membranous organelle reorganization that leads to the establishment of the preAC.

Dynamin contributes to clathrin-mediated vesicle scission at the plasma membrane (PM) [22,62,63] and the TGN [64]. Thus, to explore whether the observed effect of dynamin inhibitors is related to clathrin functions, we also tested the impact of Pitstop 2, an inhibitor of clathrin assembly [63]. This inhibitor was used to examine the role of clathrin in the AC of HCMV infected cells [48]. Although it did not inhibit HCMV AC formation, it has retarded the growth of the laboratory HCMV strains [48]. Similar to that observation in HCMV infection, Pitstop 2 did not significantly affect MCMV infected cells, neither on the pericentriolar Rab10 accumulation in the preAC at 16 hpi nor in the AC at 40 hpi (Table 2). However, Pitstop inhibited the accumulation of M55/gB in the oAC (Table 2 and Appendix A). Given that we observed accumulation of M55 at the PM of Pitstop 2-treated cells (Appendix A), clathrin inhibition may block the cycle of M55 delivery to the AC via the endocytic pathway, as shown for envelope glycoproteins in HSV-1 infection [47]. Consistent with this possibility is also an observation of M55 at the PM membrane of untreated, dynasore-treated, and MiTMAB-treated (30 µM, data not shown) cells (arrowheads in Figure 3 and Appendix A).

Disturbance of cholesterol homeostasis, especially labile cholesterol in the PM, is an important side-effect of dynamin inhibition [65,66]. Therefore, we have extracted plasma membrane labile cholesterol with methyl-β-cyclodextrin (mβCD) and explored its effect on establishing the preAC and the development of the AC. Given that mβCD is not a short-term inhibitor [67], we treated cells immediately after infection. The results presented in Appendix A and Table 2 demonstrate that this inhibitor had no significant effect, neither on Rab10 accumulation in the inner preAC and AC nor on M55 accumulation in AC (Appendix A and Table 2).

Therefore, we can conclude that preAC establishment in the early phase and AC development in the late phase of MCMV infection requires the activity of dynamin. These processes could be partially clathrin-dependent, especially M55 accumulation in oAC, but do not depend on labile cholesterol.

### 3.4. The Dynasore Treatment Affects Pericentriolar Recruitment of EE-ERC-Derived Organelles and Reorganization of the Entire Golgi Stack in the Early Phase of MCMV Infection

Using Rab10 as the primary indicator in previous experiments, we demonstrated inhibition of preAC and AC formation after treatment with dynamin inhibitors. We used Rab10 because of the plain contrast in expression between uninfected and infected cells and its early recruitment at the pericentriolar area of infected cells. However, the establishment of the preAC is a complex process that involves the rearrangement of several intracellular compartments [6,8,10,14], especially membrane intermediates at the EE-ERC-TGN interface [10]. Thus, we extended our analysis of the dynasore effect on EEs, the ERC, and the Golgi, using cellular markers that characterize specific domains in their maturation.

First, we focused on the most important steps in the maturation of the EE vacuolar domain. We analyzed recruitment of Rab5a, a marker of all stages of maturing EEs [68], and its well-known effectors: Vps34, EEA1, and Hrs. Vps34 is a Class III phosphatidylinositol-3-kinase (PI3K), generates phosphatidylinositol-3-phosphate (PI3P) at EEs [69], EEA1 binds to a PI3P and plays a role in homo- and heterotypic fusion of EEs [70] and Hrs sorts cargo destined for degradation and mobilize the ESCRT-0 (The Endosomal Sorting Complexes Required for Transport) complex to initiate the formation of intraluminal vesicles (ILV) at vacuolar EEs [71]. All these markers accumulate in iAC of CMV-infected cells [6,8,10].

Early endosomal membranes considered to localize in the preAC, were determined according to the scheme in Figure 4C. As demonstrated in Figure 4 and Table 3, at 6 hpi, ~50% of cells displayed Rab5a accumulation, whereas, at 14 hpi, it was present in ~70% of infected cells.

The accumulation of its effectors (Vps34, EEA1, and Hrs) was apparent in 20–40% of infected cells at 6 hpi, likely due to the temporal shift in the ‘over-recruitment’ and their concentration in the perinuclear area (Figure 4). Namely, Rab5a was highly recruited at EE organelles before infection, EEA1 was mainly found on dispersed vesicles, whereas Vps34 and Hrs were hardly detectable on endosomes in uninfected cells (Appendix A). Interestingly, EEA1 membranes have been less concentrated, especially 6 hpi. Although their preAC vesicles were restricted ro α < 90°, they were found on the broader area (Figure 4A,C). At 14 hpi, all markers displayed perinuclear ‘over-recruitment’ in a similar percentage of infected cells as Rab5a (Figure 4). The dynasore treatment at 4 hpi prevented perinuclear accumulation of Rab5 endosomes and ‘over-recruitment’ of its effectors, indicating that dynamin may play a role in reorganizing the vacuolar domain of EEs in the early phase infection (Figure 4 and Table 3). Even more, we observed that dynasore could induce partial dispersion and vacuolization of Rab5, EEA1, and Hrs positive endosomes in uninfected cells (Appendix A).

We next examined the effect of dynasore on the maturation of the EE recycling domain and EE-to-ERC trafficking. Besides Rab10, a marker of EE-ERC intermediates, we monitored the perinuclear accumulation of two ERC markers, Rab11 and Arf6. Rab11 typically localizes in the juxtanuclear ERC and regulates slow recycling of clathrin-dependent cargo to the PM [16,72], whereas Arf6 localizes at tubular endosomes that emanate from the ERC [73] and regulate recycling of clathrin-independent cargo [74]. As shown before, Rab10 accumulated in the pericentriolar area of infected cells, and this accumulation was inhibited with dynasore (Figure 5). Although typically found in the pericentriolar area of uninfected cells (Appendix A), the Rab11 compartment was even more concentrated after MCMV infection (64.1 ± 4.3% at 6 hpi and 78.6 ± 3.7% at 14 hpi). The dynasore treatment transiently inhibited the pericentriolar Rab11 accumulation, as demonstrated with 15.9 ± 3.9% of cells at 6 hpi and 51.29 ± 7.5% of cells 14 hpi (Figure 5 and Table 3). Interestingly, the dynasore had a similar effect on the perinuclear localization of Rab11 membranes in uninfected cells (Appendix A). These data suggest that the dynasore treatment impacts maturation of Rab11 compartments, including delivery of Rab11 membranes to the ERC, in both uninfected and MCMV-infected cells. We readily observed Rab11-positive subplasmalemmal vesicles in infected cells at 14 hpi, but rarely in dynasore-treated cells (Figure 5A, arrowheads), suggesting that dynasore impacts trafficking of Rab11 recycling carriers and their delivery to the PM [75].

The effects of dynasore on the ERC were further confirmed by analyzing Arf6 ‘over- recruitment’ in MCMV infected cells that label ERC membranes different than Rab11 [76]. Very little juxtanuclear accumulation of Arf6 was observed in uninfected cells [10] and a small fraction of 6 h-infected cells (Figure 5A), whereas Arf6 pericentriolar accumulation was present in ~70% of 14 h-infected cells (Figure 5 and Table 3). However, the dynasore treatment almost completely abolished Arf6 accumulation at 14 hpi (Figure 5 and Table 3). These data confirm that dynamin is required to reorganize both ERC domains within the inner preAC of MCMV infected cells.

Altogether, these results confirm that the dynamin contributes to the pericentriolar EE-ERC membranes accumulation and thereby in the establishment of the i-preAC.

### 3.5. Dynasore Affects Localization of Entire Golgi

Our previous data (Figure 2) demonstrated the impact of dynasore treatment on the reorganization of the Golgi. We used GM130, a protein associated with the cis-Golgi cisternae, as a marker of the Golgi reorganization events. However, our previous study demonstrated that the Golgi reorganization in the AC is more complex, especially at the Golgi entry and exit side [10]. Thus, to get insight into the effect of dynasore on the Golgi reorganization, we also examined its impact on these two sides of the Golgi using two markers, Grasp65 and Golgin 97. Grasp65 is a marker of ERES (endoplasmic reticulum exit site) and cis-Golgi, and Golgin 97 is a marker of trans-Golgi and trans-Golgi network [77,78,79].

In untreated cells, we expected cis- and medial- Golgi markers in the oAC and the TGN marker in the iAC and peripheral area of MCMV infected cells [10]. At 6 hpi, all three markers displayed reorganization of the Golgi towards the ring-like phenotype. In contrast, at 14 hpi, GM130 and Grasp65 showed a ring-like pattern, whereas Golgin 97 additionally displayed structures within the inner preAC and the cell periphery (Figure 6).

These data suggest a complex Golgi reorganization that undergoes through the early phase of infection and leads to the rotation and displacement of the peripheral Golgi elements. Dynasore treatment affected the Golgi reorganization sequence already at the early stage, preventing the dislocation of all three Golgi parts into the ring-like pattern at 6 hpi (Figure 6B). The same was observed at 14 hpi; however, the entry-side and the exit-side Golgi elements displayed a completely different appearance in dynasore treated cells. The Grasp65-positive elements were dispersed to the peripheral cytoplasm, consistent with their dislocation to the ER. By contrast, Golgin 97-positive elements were displaced from the GM130-positive ring, accumulated at the cell periphery, and did not accumulate in the pericentriolar area. This effect is consistent with the inhibitory effect of the dynasore on the establishment of the i-preAC and suggests that dynasore does not affect the displacement of Golgin97-positive membranes to the cell periphery. Altogether these data indicate that dynasore affects the Golgi events at the interface between the Golgi and the Golgi linker compartments (ERGIC and the TGN) and that Golgi reorganization into the preAC of MCMV-infected cell is a dynamin-dependent process.

### 3.6. Dynasore Does Not Affect Late Endosomes

Although CMV remodels a significant number of intracellular membranes to form AC, late endosomes and lysosomes are not involved in that process [6,10,80]. Therefore, we tested if dynasore would change the localization of Lamp1, a bona fide marker of late endosomes and lysosomes [81]. As expected, Lamp1 was found in vesicles that surrounded the pericentriolar area of fibroblasts in the early phase of MCMV infection (Appendix A). When the cells were infected in the presence of an inhibitor, the Lamp1 localization remained similar, but vesicles were more vacuolized. The vacuolization was also visualized in dynasore-treated uninfected cells (Appendix A), indicating the possible effect of dynasore on ILV formation or detaching of vesicles in late endosomal recycling [82].

### 3.7. Dynasore Prevents AC Formation When Present Only in the Early Phase of MCMV Infection

Our previous results demonstrated that dynasore added at 4 hpi prevents the establishment of the preAC. To identify when the critical events in the course of AC establishment are affected by dynasore, we performed a series of different treatment protocols with dynasore (Figure 7A). We monitored the perinuclear accumulation of Rab10 at 24 and 48 hpi and utilized the advantage of dynasore as a reversible inhibitor [33,41].

As expected, Rab10 was highly over recruited in the perinuclear area at 24 hpi and 48 hpi (Figure 7B(a)). This ‘over-recruitment’ was prevented by the presence of dynasore from the fourth hour after infection (Figure 7B(b)). The same inhibitory result was achieved when dynasore was present from the 4th to 12th hour after infection (Figure 7B(c)), suggesting that washout of the dynasore after an early-phase check-point cannot reestablish the sequence of membranous organelle reorganization required for preAC establishment. However, Rab10 accumulation in the pericentriolar area proceeded when cells were treated with dynasore during 4–6 hpi (Figure 7B(d)), confirming the reversibility of the inhibitor. These results indicate that critical processes in AC formation occur during the early phase of MCMV infection and that these processes require dynamin.

### 3.8. Dynasore Treatment in the Early Phase of Infection Blocks the Synthesis of Late MCMV Proteins and Inhibits the Release of Infective Virions

Previous experiments suggested that dynasore did not impede IE1 expression over the MCMV replication cycle (Figure 1, Figure 2, Figure 3, Figure 4, Figure 5, Figure 6 and Figure 7), consistent with the result observed on dynasore-treated HCMV-infected cells [48]. To test whether dynasore affects the early phase of infection progression, we analyzed the expression level of E1 and M57 proteins after dynasore treatment. The expression of E1 protein, expressed significantly before the 4th hour of infection [83] remained unchanged (Figure 8A). In contrast, the expression level of M57, an early protein expressed at 5–6 hpi [83], was significantly reduced at later stages of infection (Figure 8B). This effect indicates that dynasore treatment may affect the MCMV gene expression program, as suggested by the immunofluorescence analysis of M55 presented in Figure 3. Therefore, we further tested more extensively the effect of dynasore on the expression of late MCMV proteins.

As demonstrated in Figure 9A, the dynasore treatment at 4 hpi almost completely abolished the expression of M74 (glycoprotein gO). This viral structural glycoprotein is encoded by a true late gene expressed at later stages of MCMV infection, between 24 and 48 hpi [84].

Conclusions gained by the M57 and M74 protein expression analysis was confirmed by the analysis of M25 gene expression. M25 gene encodes a virion-associated tegument protein [85,86] with a complex intracellular expression pattern. M25 gene is expressed in the early phase of infection as 105 kDa protein with similar expression kinetics as M57, and as 130 kDa protein expressed as true late [87,88]. Thus, 105 kDa band was observed at 6 hpi, like M57, and 130 kDa band was fully present at 48 hpi, like M74 (Figure 9B). The dynasore treatment reduced 105 kDa M25 and abolished the expression of 130 kDa M25 (Figure 9B).

As observed for M74, the dynasore treatment also abolished the expression of M55 (glycoprotein B) (Figure 3 and Figure 9C,D). This protein is expressed in two forms in untreated cells (Figure 9B), the large 130 kDa form, which serves as a precursor [61], and a cleavage product of 55 kDa [89,90]. The dynasore treatment at 4 hpi abolished the expression of unprocessed M55 protein (130 kDa) and the cleaved M55 fragment (55 kDa) (Figure 9C). These data are consistent with the observation of inhibited MCMV late protein (M55/gB) expression in triple dynamin knock-out cells [38]. Interestingly, the same result was achieved when cells were incubated with dynasore only during the period of 4–14 h after infection. (Figure 9D). This result was also confirmed by the flow cytometric detection of M55 in MCMV-infected cells (Figure 9E), demonstrating that the effect of dynasore in the early phase of infection is irreversible, although the dynasore itself is a reversible inhibitor, as shown in Figure 7.

Altogether, protein expression analysis demonstrated that dynasore inhibits the expression of MCMV proteins in the late phase of infection and abolishes late gene expression. Thus, it was reasonable to expect that dynasore treatment will substantially abolish infectious virion production. To test this, we monitored infectious virions released in the supernatant by the standard plaque assay. As demonstrated in Figure 9F, both continuous (from the 4 hpi to 48 hpi and from 4 hpi to 72 hpi) and the transient (from 4 to 14 hpi) dynasore treatment significantly inhibited infectious virion production. In 4–14 hpi dynasore-treated cells, the release of infectious virions at 48 hpi was reduced for 7.6 fold, which was even more reduction observed in 4–48 treated cells (55 fold). A similar decrease of infectious virion release was observed after 72 h in 14–4 hpi and 4–72 hpi dynasore-treated cells (33.8 fold versus 18 fold, respectively).

Together, these results indicate that dynasore-sensitive processes in the early phase of infection are crucial for establishing the preAC and the expression of late-phase MCMV proteins.

## 4. Discussion

This study used dynasore, a rapidly acting inhibitor of the dynamin GTPase activity [41], to explore the role of dynamin in the membranous organelle reorganization during the early phase of CMV infection. This reorganization establishes a basic configuration of the structure known as the assembly compartment (AC). In this study, we denoted this configuration as the preAC since the full configuration of the AC is established in the late phase of infection, after the expression of more than half of CMV genes [83]. Through the temporal analysis of cis-Golgi remodeling, we identified Golgi unlinking and fragmentation as the first events present at 4 hpi. The initial events are followed by a sequence of Golgi reorganization that ends up with the displacement of the Golgi into the ring-like structure, which is observed in 30% of cells at 6 hpi under infection conditions used in this study. This structure is gradually established in 70–80% of infected cells and maintained throughout the preAC development and later in fully established AC. The establishment of the ring-like Golgi phenotype coincided with the expansion of EE-, ERC-, and TGN-derived membranous elements within the Golgi ring. This expansion, as monitored by “over-recruitment” of Rab10 in the pericentriolar area of infected cells, was identified at 5–6 hpi. The dynasore treatment at 4 hpi prevented both the Golgi displacement and Rab10 accumulation. A similar effect was achieved with other dynamin inhibitors, but not with the clathrin and cholesterol inhibitors. These data indicate that dynamin plays a crucial role in establishing the preAC and that this role is not associated with clathrin vesicle scission and cholesterol depletion. The dynamin role is required at the initial stages of preAC establishment and cannot be reversed by aborting dynamin inhibition at later stages of the early phase of infection. As expected, dynasore treatment strongly inhibited infectious virion release; however, this effect cannot be attributed only to the absence of the AC since the dynasore treatment also inhibited the expression of late viral proteins.

The role of dynamin in CMV assembly and egress has been addressed in two recent studies. In the HCMV study [48], the dynamin role was also assessed using dynasore, whereas in the MCMV study [38] by monitoring virion production in the triple dynamin knock-out cells. Both studies agreed with the essential role of dynamin in virion production and suggested that its activity may be related to the establishment of the AC. Our study confirms the indispensable role of dynamin in MCMV production and demonstrates that dynamin plays an essential role in the earliest stages of the AC biogenesis, at a stage of the establishment of preAC in the early phase of infection. Although in the previous studies [38,48] the role of dynamin in the AC biogenesis was not studied in detail, these results, together with the results of our study, indicate that the disruption of dynamin-associate functions may be a crucial step targeted by beta-herpesviruses to initiate a complex set of events that lead to the establishment of the structure known as the AC.

The AC develops after massive rearrangement of endosomal and secretory membranes in HCMV and MCMV infected cells [6,7,8,9,10,11,80,91,92,93]. These rearrangements follow a similar, but not identical, sequence of events that displace the Golgi and expand the Golgi-linked membranous system within the Golgi ring. Furthermore, in MCMV infected fibroblasts, this sequence is faster than in HCMV infected cells under cell culture conditions. Our recent research that characterized these reorganizations using 64 markers of membranous organelle maturation showed that many of these events involve reorganization of the EE, ERC, and TGN suggesting dysregulation of the interface between these organelles. This dysregulation is manifested as ‘over-recruitment’ and accumulation of membranous elements of the EE-ERC-TGN interface, including several small GTPases from the Rab and Arf subfamily [10].

Given that it is widely accepted that these GTPases are crucial players in defining the identity of membranous domains, the ‘over-recruitment’ of some of them, which are not highly detectable in uninfected cells, indicate dysregulation of Rab and Arf cascades at the membrane of infected cells. One of these GTPases is Rab10, which is transiently recruited in short-lived EE-ERC intermediates in uninfected cells [54,55], resulting in low endogenous visualization at membranous organelles. However, Rab10 is highly recruited at pericentriolar membranes in infected cells, thereby displaying the reorganization events that occur within the Golgi ring. In this study, we used the Rab10 ‘over-recruitment’ to monitor the effect of dynasore and demonstrate that the dynamin activity is required for reorganization events at the EE-ERC interface. However, we also extended our analysis to the set of EE-ERC markers to show that dynasore acts also on EE- and the ERC-associated events, and by the use of Golgin97 that also acts on the TGN contribution to the inner membranous agglomerate of the preAC. The analysis of EE-ERC-TGN markers indicates that dynamin may contribute at multiple and spatially and temporary distinct regulatory cascades within the inner preAC configuration.

In addition to the extensive reorganization of the EE-, ERC-, and TGN-derived organelles, this study also addressed reorganization of the Golgi, which coincided with the restructuring of the EE-ERC-TGN interface. We focused on the cis-Golgi membranes that can be visualized by the well-defined marker GM130 and explored the entry and exit sites of the Golgi by using Grasp65 and Golgin 97, respectively. Our study demonstrated that the Golgi reorganization may proceed with the restructuring of the EE-ERC-TGN interface and may be initial events in establishing the preAC configuration. Using dynasore, we showed that dynamin might be essential also for the Golgi reorganization events. The study reveals that dynamin inhibition affects not only the repositioning of the cis-Golgi but also the entry (ERGIC) and exit (TGN) side of the Golgi. These effects of dynamin inhibitors reveal more complexity and suggest that more thorough studies are required to get a complete picture of the Golgi reorganization during the establishment of the preAC.

The Golgi reorganization events in MCMV infected cells seem similar to those observed in HCMV infected cells [56]. The HCMV study demonstrated that the Golgi fragmentation is an integral part of AC development. Golgi fragmentation indicates activation of unlinking processes both in HCMV [56] and MCMV infected cells. The assembly of Golgi stacks in healthy cells is maintained by a family of Golgi proteins, named golgins, but Golgi unlinking and fragmentation normally occur during mitosis and cell motility [58,77]. The HCMV study suggested that Grasp65 phosphorylation, which is essential for Golgi fragmentation and oAC formation in HCMV infected fibroblasts, probably involves mitotic kinases [56]. Furthermore, it has been shown that the pseudomitotic state is maintained in the HCMV infection [94]. Our study also points to the Golgi unlinking as an initial event and demonstrates that the Golgi undergo a sequence of transformations at 4–6 hpi, including unlinking, fragmentation, compacting, and dislocation. This sequence is abolished in cells treated with dynamin inhibitors, indicating that dynamin activity is required in the Golgi reorganization sequence.

The inhibitory effect of dynasore on the EE-, ERC-, and TGN-derived organelle reorganization within the inner preAC and the Golgi repositioning in the outer preAC indicate the essential contribution of dynamin in the establishment of the entire preAC. Thus, it is intriguing how inhibition of one protein may result in such a massive membranous organelle reorganization? It is known that dynamin contributes to several processes within the cell: (i) scission of endocytic vesicles and endosomal tubules [21,22,23], (ii) actin dynamics [29,30], (iii) centrosome binding and regulation of the cell cycle [32], and (iv) microtubule instability and regulation of membrane traffic [28,31]. The scission function of dynamin is essential for endocytic processes [21,22,23], the development of recycling carriers at EEs [33,37,95,96], and outgoing carriers at the TGN [24]. Even more, the dynamin over activation may cause Golgi vesiculation [24,36]. Interestingly, recently published research has shown that infection of primary neurons with HSV-1 stimulates Src kinase, followed by dynamin activation and consequent Golgi fragmentation [97]. Thus, dynamin over recruitment and its membrane scission function may explain some events that characterize preAC establishment. However, although over recruited within the late inner AC (at 48 hpi), at early stages, dynamin was found in the outer preAC, and the extent of its presence after immunofluorescence staining can be hardly characterized as “over-recruitment” [10]. Therefore, the complete sequence of events leading to the establishment of the preAC can hardly be explained by over-activating dynamin, especially the heterogeneous processes that simultaneously occur in the inner preAC. Furthermore, dynasore can also affect some dynamin-independent functions related to the disruption of cholesterol homeostasis [66]. However, the depletion of labile cholesterol with methyl-β-cyclodextrin did not prevent the establishment of the preAC and the AC, indicating that the dynamin-independent effects of dynasore are unlikely (Appendix A).

Another possible dynamin function in CMV infection is its role in regulating the dynamic instability of microtubules required for normal perinuclear trafficking of TfR into pericentriolar recycling endosomes and establishment of mature cisternal Golgi [31]. In line with this, nocodazole treatment has also been shown to disintegrate AC in HCMV infected cells [98]. Moreover, it has recently been demonstrated that HCMV infection converts the Golgi part of the AC to act as a microtubule-organizing center (MTOC) which generates rapidly acetylated and stabilized microtubules by activation of the end binding protein 3 (EB3) [99]. This process enables nuclear rotation to get the best position to transport mature nucleocapsids in front of the AC. Furthermore, both EB1 and EB3 proteins are essential for AC integrity. The EB1 depletion dispersed the Golgi [99], as did the dynasore treatment in our experiments. Interestingly, EB1 binding to the plus end of dynamic microtubules was significantly reduced in dynamin-2 depleted cells [31]. Thus, dynamin may contribute to the microtubule dynamic at the Golgi membranes in the early phase of infection, destabilizing Golgi and its homeostatic feedback with linker compartments, ERGIC, ERC, and the TGN [58]. The destabilization of this feedback may reconfigure these compartments that relocate within the dislocated Golgi ring. Treatment with the dynasore during the beginning of preAC establishment but not at later stages is sufficient to prevent AC formation, as demonstrated in our kinetic experiments. This suggests a sequence of membranous system reorganization aligned with MCMV genes expression. The disruption of this sequence establishes the configuration of the membranous system that disables compensation of the skipped step.

The role of dynamin has been studied in infections with many viruses, as reviewed in Harper et al. [39]. Its function in the endocytic entry represents the most exploited dynamin-dependent mechanism. However, some viruses have been demonstrated to mobilize dynamin for post-entry processes, such as forming the compartments for virion assembly, recycling of envelope glycoproteins, or virus egress [36,38,46,47,48,100]. Dynamin has also been shown to play an important role in HSV-1 replication by its contribution to the endocytosis-dependent delivery of envelope glycoproteins to the assembly site for the production of new virions [47,101]. This type of cycling has not been demonstrated in CMV, but it can be a general mechanism for concentrating all components required for virion assembly. In line with this are our observation of M55/gB at the PM of untreated cells and its retention in cells treated with dynamin inhibitors as dynasore or MiTMAB and clathrin inhibitor Pitstop (Figure 3A). Therefore, these results suggest dynamin-dependent M55/gB trafficking to the AC using the TGN-PM-EE-AC trafficking route, similar to the trafficking of furin [102], the endoprotease that cleaves M55/gB protein. Although it has been shown gB cleavage is indispensable for in vitro HCMV replication [90], it would be interesting to investigate their endocytic traveling.

Results of our study indicate that some of the critical events at the beginning of preAC establishment require the activity of dynamin. These events are needed to reconfigure further the membranous system that continues in the early and late phases of infection. In this study, we took advantage of the rapidity and reversibility of dynamin inhibitors to intervene at the proper stage of preAC establishment. Delayed intervention would not stop the process, and abortion of the intervention at later stages, would not regenerate it. Thus, it is challenging to see whether the long-term intervention by the dynamin knock-down or expression of dominant-negative dynamin mutants could clarify this critical step in the preAC biogenesis. Long-term interventions into host-cell functions constrain clarification of their role during viral infection. For example, the knock-down of all three dynamin genes drastically prolonged the MCMV replication cycle [38]. Similarly, knock-down of host cell function that contributes at the earlier stages, such as Arf1, disables assessing its role in the preAC biogenesis, although it is highly over recruited to the membranes of the inner preAC [103]. Thus, a combination of genetic tools and rapidly acting inhibitors is essential in the studies of preAC biogenesis to build the complete picture of such a complex process.

## 5. Conclusions

Dynamin-regulated cellular processes are manipulated by different pathogens, including viruses, to achieve their primary goals—replication and survival. Our study suggest that dynamin plays an essential role in the early phase of MCMV assembly compartment (preAC) establishment and that these early processes are crucial for virus maturation and egress. The final goal of all studies investigating molecular mechanisms of viral replication cycles, including the development of viral assembly compartments, is to accumulate a satisfactory amount of knowledge for the future development of more specific antiviral drugs. Dynamin inhibitors are primarily investigated for future medical treatments in cancer therapy [49,50]. However, while still not applicable for the treatment of CMV infections, they can help us clarify some essential processes in CMV pathogenesis.

## Figures and Tables

**Figure 1 life-11-00876-f001:**
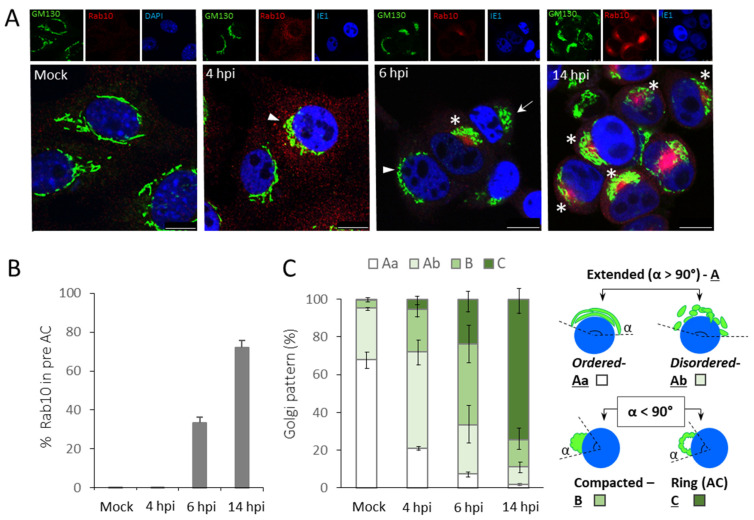
Stages of preAC development in MCMV infected Balb3T3 fibroblasts. Cells were plated on coverslips and infected with Δm138-MCMV (MOI 10). (**A**) Rab10 (EE/ERC marker; red), GM130 (cis-Golgi; green), and IE1 (a marker of infection; blue) were visualized at indicated times during the early phase of infection. In mock-infected cells, the blue color indicates DAPI. Arrowhead indicates disordered Golgi (“Ab”), arrow indicates compacted Golgi (“B”), and asterisk indicates ring-shaped Golgi (“C”). Scale bar is equal to 10 µm. Representative focal-plane images of three independent experiments are shown. (**B**) The percentage of infected cells with Rab10 in preAC. (**C**) The diagram shows the ratio of different Golgi patterns visualized by GM130. The definition of different patterns is schematically presented in the right panel. Details are explained in the main text. Mean values are plotted, and error bars show standard deviations.

**Figure 2 life-11-00876-f002:**
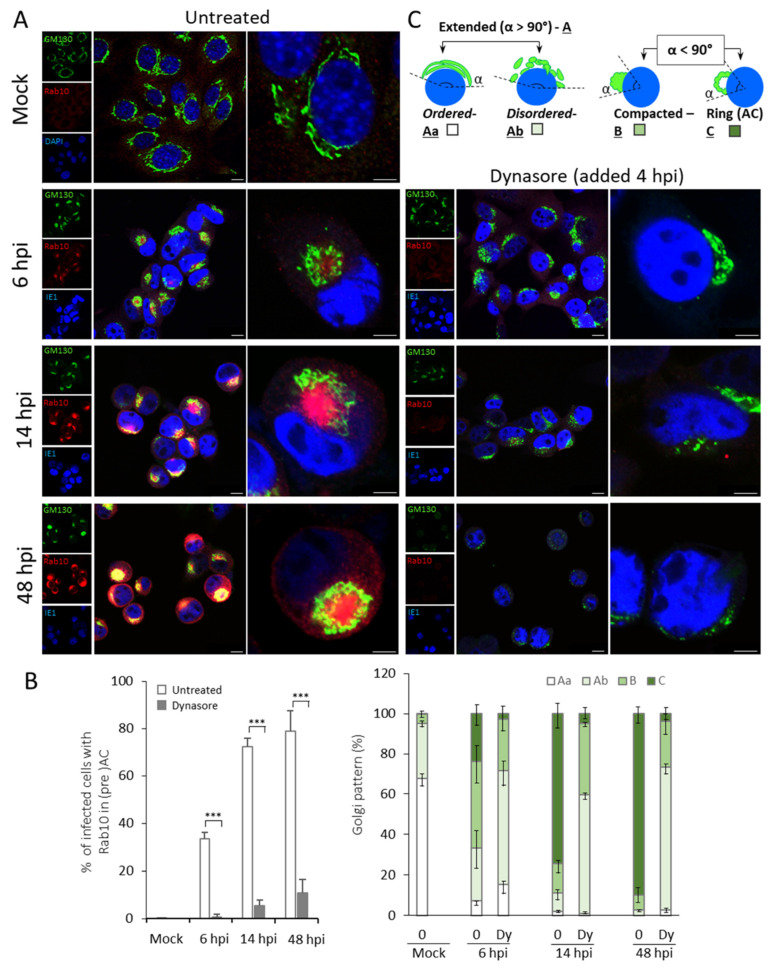
Dynasore inhibits the development of preAC. (**A**) Cells were plated on coverslips, infected with Δm138-MCMV, and at 4 hpi treated with dynasore (80 µM) or left untreated. At 6, 14, and 48 hpi, the cells were fixed, permeabilized, and stained against Rab10 (EE/ERC marker; red fluorescence), GM130 (cis-Golgi; green fluorescence), and IE1 (a nuclear marker of infection; blue fluorescence). In mock-infected cells, the blue color represents the DAPI nuclear staining. Scale bar is equal to 10 µm in smaller and 5 µm in larger magnifications. Representative images of three independent experiments are shown. (**B**) Quantification of Rab10 accumulation and the Golgi patterns in immunofluorescence experiments. The left panel presents the percentage of infected cells with juxtanuclear accumulation of Rab10 (inner preAC), whereas the right panel presents ratios of different Golgi patterns visualized by GM130. Data represent the mean values, and error bars show standard deviations. The significance to the untreated samples of the same kinetics was determined by the Student’s *t*-test (*** *p* < 0.001). (**C**) Schematic presentation of Golgi patterns.

**Figure 3 life-11-00876-f003:**
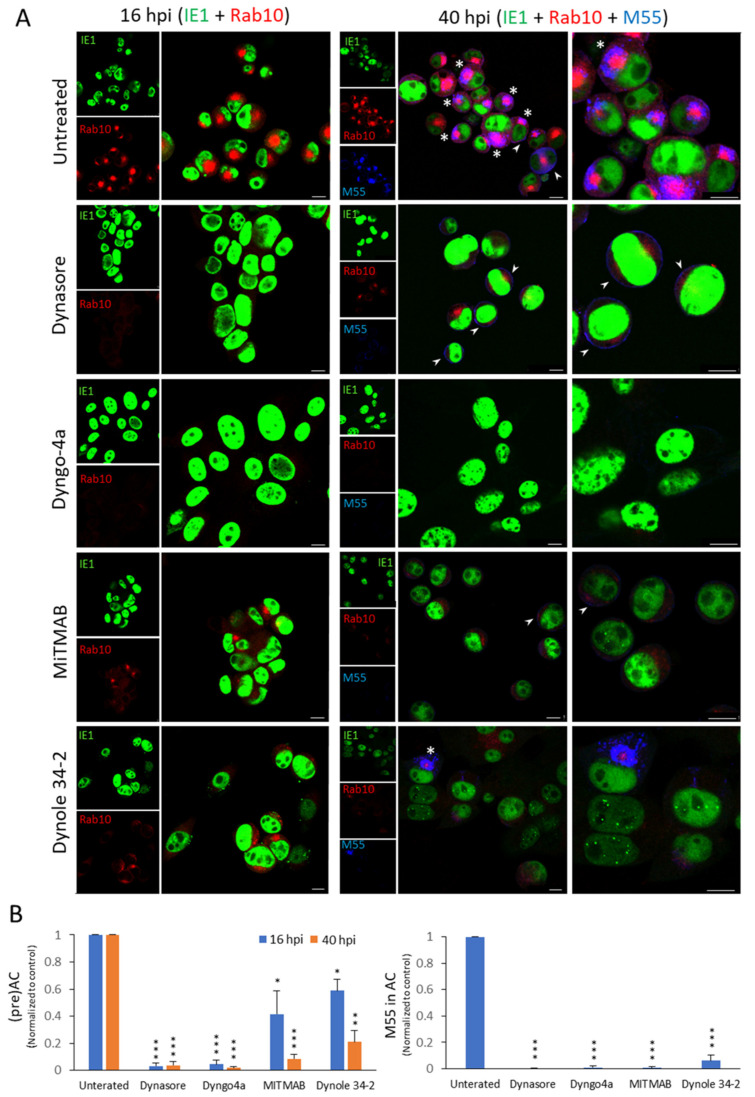
The effect of dynamin inhibitors on the establishment of the preAC and development of the AC. Δm138-MCMV infected cells were treated with dynasore (80 µM), Dyngo-4a (200 µM), MiTMAB (20 µM), and Dynole-34-2 (10 µM) at 4 hpi, and incubated in the presence of inhibitors up to 16 and 40 hpi. (**A**) Double and triple immunofluorescence images of 16 and 40 h infected cells, respectively, stained against IE1 (green), Rab10 (red), and M55/gB (blue). Asterisks point to M55 in the AC and arrowheads to M55 at the cell surface. The scale bar is equal to 10 µm. Representative images of three independent experiments are shown. (**B**) Quantification of the impact of inhibitors on establishing the preAC indicated by Rab10 accumulation (left panel) and M55 loading into the oAC (right panel). Results on graphs are presented as normalized to control [(%Rab10_Inh._/%Rab10_Ø_) or (%M55-AC_Inh._/%M55-AC_Ø_)]. Only IE1^+^ cells were analyzed. The data represent the mean ± SD from 3–4 independent experiments. The significance of differences to untreated samples from the same kinetics was determined using the Student’s *t*-test (*** *p* < 0.001, ** *p* < 0.01, * *p* < 0.05).

**Figure 4 life-11-00876-f004:**
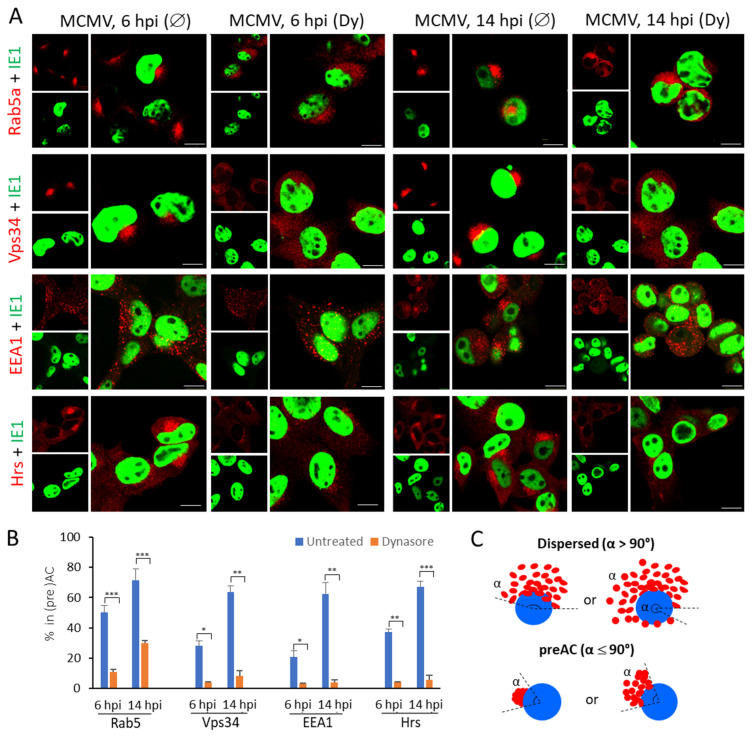
The effect of dynasore on the pericentriolar accumulation of EE in MCMV infected Balb3T3 cells. Cells were plated on coverslips, infected with Δm138-MCMV, and treated with dynasore (Dy, 80µM) at 4 hpi or left untreated (Ø). At 6 and 14 hpi, cells were proceeded for immunofluorescence. (**A**) Immunofluorescence images of EE markers (red) and IE1 (green) at 6 and 14 hpi. (**B**) Percentage of (IE1^+^) cells demonstrating pericentriolar accumulation of EE markers. (**C**) The schematic presentation of localization of EE vesicles in preAC. Mean values from 2–4 independent experiments are plotted, error bars show standard deviations. The significance of differences to untreated sample of the same kinetics was determined using the Student’s *t*-test (*** *p* < 0.001, ** *p* < 0.01, * *p* < 0.05).

**Figure 5 life-11-00876-f005:**
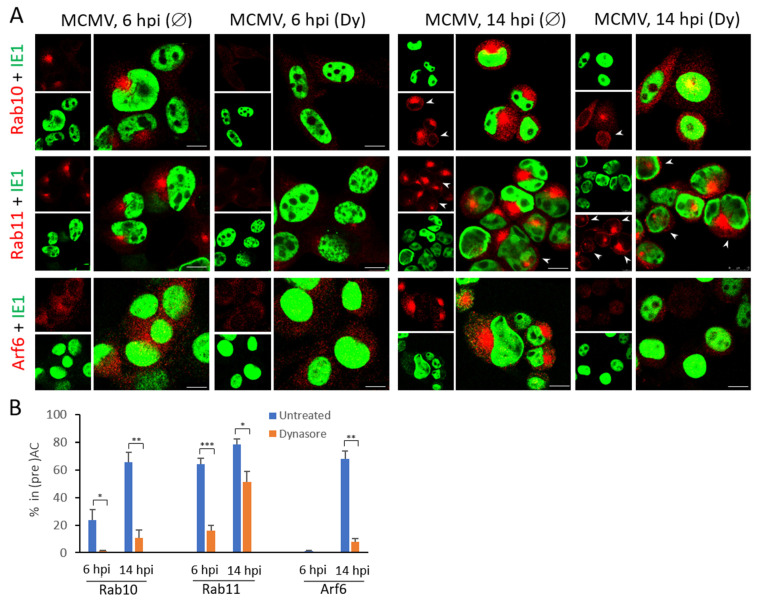
The effect of dynasore on the ERC accumulation in MCMV infected cells. Cells were plated on coverslips, infected with Δm138-MCMV, and treated with dynasore (Dy, 80 µM) at 4 hpi or left untreated (Ø). At 6 and 14 hpi, the cells were proceeded for immunofluorescence. (**A**) Immunofluorescence images of ERC markers (red) and IE1 (green) at 6 and 14 hpi. Arrowheads indicate peripheral Rab11^+^ vesicles. (**B**) Percentage of (IE1^+^) cells demonstrating pericentriolar accumulation of ERC markers. Mean values from 2–4 independent experiments are plotted, error bars indicate standard deviations. The significance of differences according to the untreated sample of the same kinetics was determined using the Student’s *t*-test (*** *p* < 0.001, ** *p* < 0.01, * *p* < 0.05).

**Figure 6 life-11-00876-f006:**
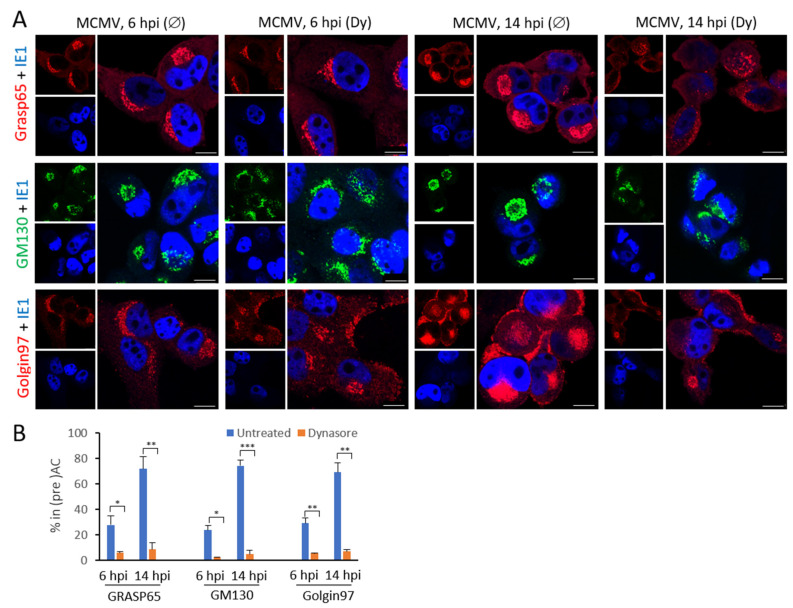
The effect of dynasore on the reorganization of the Golgi in MCMV infected cells. Δm138-MCMV infected cells were treated with dynasore (80 µM; Dy) at 4 hpi or left untreated (Ø), and at 6 and 14 hpi proceeded for immunofluorescence. (**A**) Immunofluorescence images of Grasp65 (red), GM130 (green), Golgin97 (red), and IE1 (blue) at 6 and 14 hpi. (**B**) The percentage of MCMV infected cells (IE1-positive) with the perinuclear ring-like pattern of Grasp65, GM130, and Golgin97. The data represent the mean from 3 independent experiments, and error bars show standard deviations. The significance of differences to untreated samples of the same kinetics was determined using the Student’s *t*-test (*** *p* < 0.001, ** *p* < 0.01, * *p* < 0.05).

**Figure 7 life-11-00876-f007:**
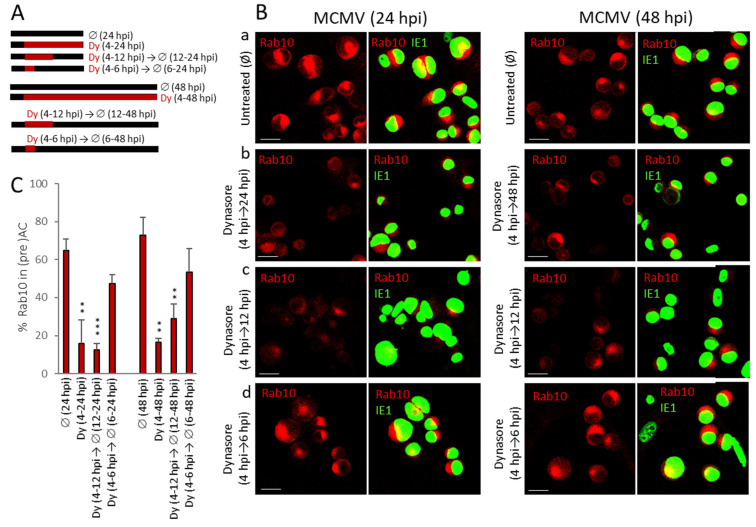
The effect of dynasore on (pre)AC development in MCMV infected Balb3T3 cells. Cells were plated on coverslips and infected with Δm138-MCMV (MOI 10) for 24 or 48 h. (**A**) Infected cells were left untreated or treated with dynasore (80 μM) according to presented pulse-chase protocols. (**B**) Rab10 (red) and IE1 (green) in different kinetics of dynasore treatments. Images from a representative experiment are shown. The scale bar is equal to 20 µm. (**C**) The percentage of infected cells with Rab10 in AC. Mean values of four independent experiments are plotted, error bars show standard deviations. The significance to the untreated sample of the same kinetics was determined using the Student’s *t*-test (*** *p* < 0.001, ** *p* < 0.01).

**Figure 8 life-11-00876-f008:**
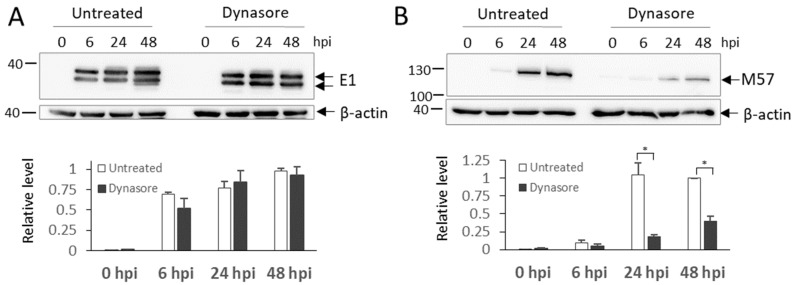
The effect of dynasore on the expression of E1 and M57 in MCMV infected cells. Δm138-MCMV infected cells were treated with dynasore (80 μM) at 4 hpi or left untreated, and at indicated times proceeded to the Western blot analysis. β-actin, a cellular loading control, was visualized sequentially after E1 (**A**) or simultaneously with M57 (**B**) at the same membrane. Representative Western blots and mean values of corresponding densitometry analysis of two independent experiments are shown. The significance to the untreated sample of the same kinetics was determined using the Student’s *t*-test (* *p* < 0.051).

**Figure 9 life-11-00876-f009:**
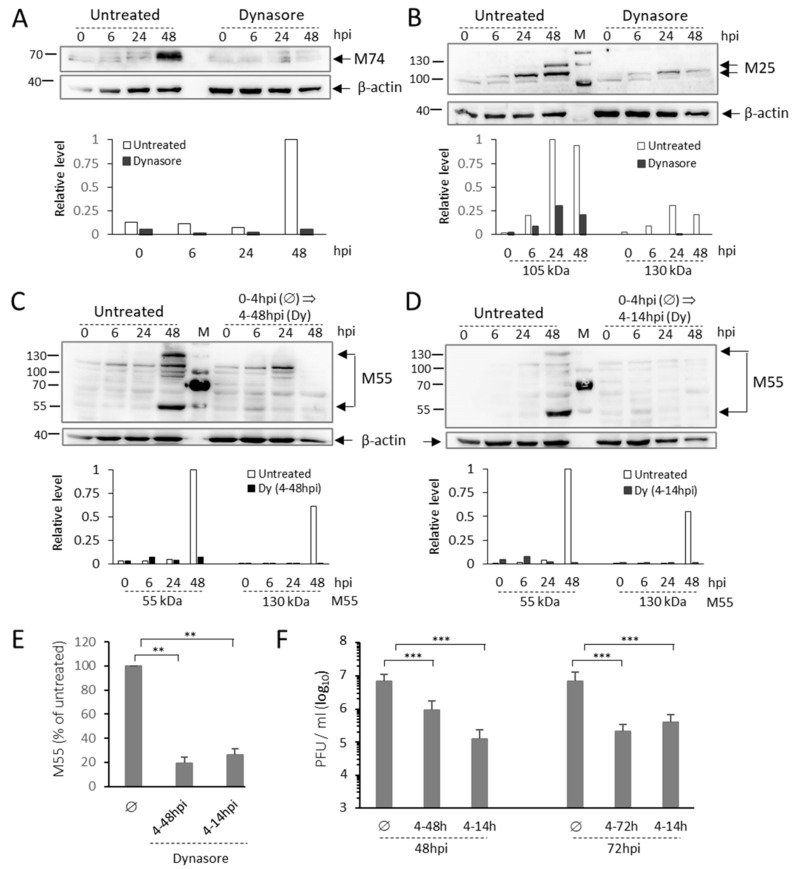
The effect of dynasore on the expression of late MCMV proteins and virion production. Δm138-MCMV infected cells were treated with dynasore (80 μM) at 4–48 hpi or 4–14 hpi. At indicated times, the samples were analyzed by Western blot (**A**–**D**), flow cytometry (**E**), and plaque assay (**F**). (**A**–**C**) Representative Western blots of M74 (**A**), M25 (**B**), and M55 (**C**) expression in the 4–48 hpi dynasore-treated cells. (**D**) Western blot of M55 expression in 4–14 hpi dynasore-treated cells. In all Western blot experiments, the β-actin, a cellular loading control, was visualized either sequentially or simultaneously at the same membrane and used to normalize the signal in densitometric analysis. (**E**) Flow cytometric quantification of M55 expression in untreated (Ø) and 4–48 or 4–14 hpi dynasore-treated cells. The data present the relative percentage of M55-positive cells (mean ± SD). (**F**) Quantification of extracellular virions in supernatants at 48 and 72 hpi of untreated and dynasore-treated (4–48, 4–72, and 4–14 hpi) cells. The data represent the mean values of four independent experiments, and error bars show standard deviations. The significance of differences was determined using the Student’s *t*-test (*** *p* < 0.001, ** *p* < 0.01).

**Table 1 life-11-00876-t001:** Quantification of pericentriolar accumulation of Rab10 and the Golgi reorganization in untreated and dynasore (Dy)-treated MCMV-infected cells.

	Pericentriolar Rab10 (%) *		Golgi Patterns (%) *							
			Aa		Ab		B		C	
	Ø	Dy	Ø	Dy	Ø	Dy	Ø	Dy	Ø	Dy
Mock	0	-	67.8 ± 4.2	-	27.2 ± 1.4	-	4.4 ± 2.3	-	0.4 ± 0.7	-
6 hpi	33.3 ± 2.9	0.6 ± 1.1	7.3 ± 1.8	15.4 ± 4.9	25.9 ± 10.5	56.1 ± 7.2	42.7 ± 10.7	25.9 ± 9.0	23.9 ± 6.3	2.6 ± 2.9
14 hpi	72.1 ± 3.6	5.4 ± 2.2	1.9 ± 0.7	0.9 ± 0.5	9.4 ± 3.9	58.7 ± 2.9	14.2 ± 4.2	35.3 ± 1.6	74.4 ± 7.5	5.0 ± 3.7
48 hpi	78.9 ± 8.5	10.7 ± 5.8	0	2.7 ± 1.3	2.8 ± 0.7	70.5 ± 3.7	7.1 ± 4.2	22.9 ± 6.2	90.0 ± 5.0	3.8 ± 3.7

* mean ± standard deviation. Δm138-MCMV infected Balb3T3 cells were treated with dynasore (80 µM) at 4 hpi (Dy) or left untreated (Ø). The cells were processed for immunofluorescence labeling at 6, 14, and 48 hpi. The percentage of cells in three independent experiments with the pericentriolar accumulation of Rab10 and different Golgi patterns was quantified as explained in Section 2.

**Table 2 life-11-00876-t002:** Quantification of preAC establishment and AC formation in MCMV infected cells after inhibition of dynamin and clathrin functions or cholesterol depletion.

Inhibitor	Number of Cells		Rab10 Accumulation (%) *		M55 in the oAC (%) *		Viability *	
	16 hpi	40 hpi	16 hpi	40 hpi	16 hpi	40 hpi	16 hpi	40 hpi
Untreated	4209	2386	65.5 ± 11.5	81.0 ± 5.9	0	56.6 ± 8.6	96.2 ± 3.0	90.5 ± 4.7
Dynasore (80 μM)	3000	1757	2.2 ± 1.6	2.9 ± 2.5	0	0.1 ± 0.2	98.0 ± 2.8	92.5 ± 0.7
Dyngo 4a (200 μM)	1006	649	2.4 ± 1.5	1.3 ± 0.3	0	0.3 ± 0.5	90.0 ± 3.2	85.0 ± 2.1
MiTMAB (20 μM)	1547	935	30.0 ± 13.6	7.1 ± 3.0	0	0.3 ± 0.6	91.0 ± 4.0	81.0 ± 10.8
Dynole 34-2 (10 μM)	1078	623	41.9 ± 8.4	17.6 ± 5.0	0	4.9 ± 1.8	88.5 ± 6.4	70.5 ± 9.1
Pitstop 2 (50 μM)	1679	1652	45.8 ± 11.1	70.0 ± 9.4	0	10.5 ± 9.2	95.0 ± 2.4	89.0 ± 3.1
Methyl-β-CD (7.5 mM)	1985	1194	52.9 ± 19.2	67.5 ± 0.8	0	37.9 ± 15.9	97.0 ± 2.1	89.0 ± 1.5

* mean ± standard deviation. Δm138-MCMV infected Balb 3T3 cells were treated with the inhibitors and proceeded for immunofluorescence staining of Rab10 and M55, as explained in Figure 3 and Appendix A. The data represent the mean ± SD from 3–4 independent experiments.

**Table 3 life-11-00876-t003:** Quantification of dynasore effect on the recruitment of EE/ERC/Golgi markers in preAC of MCMV infected Balb3T3 cells.

Marker of Cellular Compartment	The Ratio of Infected Cells (%) with the Accumulated Marker in Pericentriolar Membranes *			
	6 hpi		14 hpi	
	Untreated	Dynasore	Untreated	Dynasore
Early Endosomes				
Rab5	50.1 ± 4.5	10.7 ± 1.9	71.5 ± 7.6	29.7 ± 1.6
Vps34	28.0 ± 3.5	4.0 ± 0.2	63.8 ± 4.1	8.3 ± 3.2
EEA1	20.7 ± 4.1	2.9 ± 0.6	62.4 ± 7.5	4.0 ± 1.3
Hrs	37.2 ± 1.8	4.0 ± 0.1	67.3 ± 3.6	5.6 ± 2.9
Early Endosomes to Endosomal Recycling Compartment				
Rab10	33.3 ± 2.9	0.6 ± 1.1	72.1 ± 3.6	5.4 ± 2.2
Endosomal Recycling Compartment				
Rab11	64.1 ± 4.3	15.9 ± 3.9	78.6 ± 3.7	51.3 ± 7.5
Arf6	1.0 ± 0.5	0	67.81 ± 5.8	7.6 ± 2.7
Golgi Compartments				
Grasp65 (ERGIC-cis-Golgi)	28.0 ± 2.0	6.0 ± 0.5	71.8 ± 5.1	8.7 ± 2.8
GM130 (cis-Golgi)	23.9 ± 13.3	2.6 ± 2.9	74.4 ± 7.5	5.0 ± 3.7
Golgin97 (trans-Golgi)	29.1 ± 2.0	5.4 ± 3.1	69.2 ± 6.5	7.0 ± 0.6
Grasp65 (ERGIC-cis-Golgi)	28.0 ± 2.0	6.0 ± 0.5	71.8 ± 5.1	8.7 ± 2.8
Late Endosomes				
Lamp1	0	0	0	0

* mean ± standard deviation. Balb 3T3 cells were infected with Δm138-MCMV, treated with dynasore (80 µM), or left untreated, and proceeded for immunofluorescence. Quantification was performed as explained in Section 2. The data represent mean ± SD from 2–4 independent experiments.

## Data Availability

The data presented in this study are available on request from the corresponding author.

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
