# Peer review of "Dynamin Inhibitors Prevent the Establishment of the Cytomegalovirus Assembly Compartment in the Early Phase of Infection"

_life, 2021, doi:10.3390/life11090876_

Round 1

Reviewer 1 Report

In the paper titled “Dynamin inhibitors prevent the establishment of the cytomegalovirus assembly compartment in the early phase of infection” by Štimac et al., the authors do a detailed, temporal analysis of the role that the host protein dynamin may play in the early phases of MCMV infection. The study approach was primarily cell culture infection with well-established inhibitors of dynamin using fluorescence microscopy and protein analysis to chart the formation of the viral assembly complex. While this study is not the first to explore the role of dynamin in CMV infection, the authors do extend the work by exploring the intracellular dynamics at a higher resolution and in finer detail. The exploration of multiple aspects of the Golgi structures is a particularly nice addition to the literature.

Overall, I have to commend the authors for putting together a thoughtful, thorough, interesting manuscript. The storyline is well ordered and explained in a clear way; the figures and tables are extremely well composed; and the findings are important for investigators of CMV as well as those who work on intracellular trafficking. Below are some specific comments on the manuscript.

Specific comments:

I am glad that the researchers included multiple inhibitors of dynamin in the analysis. This is a strength. Even though all of these inhibitors have been well-established in the literature, the fact is that all inhibitors have unique dynamics and the risk of off-effects, and so the bolstering of dynamin inhibition was a good move.

I’m glad that the authors included dynasore-treated uninfected cells in the supplementary materials. However, Figure S1 is a little confusing. There is labeling of “untreated” in the images, but then treated cells at 2, 10, and 40 hours. Were the “untreated” cells really 0-hour treatment? In other words, did you add the virus and the inhibitor and then analyze at a zero timepoint? If so, then “untreated” should be changed to 0-hours, or the table needs to be updated. The left column of the table refers to time, but the images refer to treatment and time. I hope that made sense. Also regarding Figure S1, the panel says 44 hours and the table says 40 hours. A small disconnect, I know – but accuracy will improve the figure.

Note that in Figure S2, the title of the figure suggests that the cells were pre-treated with dynasore, but they were actually treated alongside infection. Rather than saying that they were treated before infection, I would say that they were treated before viral uptake or viral internalization. I think that might be more accurate. I believe that the authors may have been using the term “infection” and “uptake” synonymously? Please confirm.

I’m glad that the authors did multiple analyses of the EE-to-ERC stage with multiple markers (Rab11, Arf6, Rab10). These were key experiments for the project.

The exposures for some of the microscopy seems a little high. I realize that this is not something that can be remedied in retrospect, and I don’t think that it is a problem for analysis, since the authors did a nice job of quantifying their results. It is certainly just an inherent problem with reporting microscopy.

The writing overall is very good, although there are some areas where the grammar is a bit awkward. The manuscript could use one last round of editing for grammar. Also, there is a mix of using commas versus decimal points in some of the numeric reporting, and that should definitely be cleaned up before publication. However, I would note that the “storytelling” in the Results is very good—I appreciate that the authors explained the significance of each of the markers they assessed along with the significance of the markers.

Regarding the cell compartment markers, I reviewed the main markers that the authors chose, and it seems that the authors did the best job of choosing well-established markers and antibodies as is possible. So I can’t really see any problems in terms of quality control. I am going on the good faith reporting of the researchers—they seem to have covered all the bases, as far as I can tell.

I don’t want to suggest unnecessary extra work for the authors, because the manuscript is so nice. But a summary schematic might be nice? Just a suggestion—not a requirement. I mention it because the authors included so many nice schematics in the figures.

A few minor comments:

The reporting of percentages out to 2 significant figures is perhaps a bit beyond the accuracy of the microscopy measurements. It’s hard to imagine 0.01 cellular at this resolution. I think that reporting out to 1 significant digit for these measurements might be sufficient – but it’s not crucial, and I know that this is often the norm in other publications. I will defer to the editor in regard to the reporting preferences of the journal for fluorescence microscopy.

The term “data” is normally plural in the scientific literature. Again, I defer to the journal’s house style.

The phrase “over-recruitment” might be easier to read if it is hyphenated. While it is not acting as an adjective, it is a bit of a coined term, so the punctuation might make it easier on the reader.

Some minor comments on Figures & Tables:

The figures and tables are laid out and sized very nicely. I am particularly grateful for the very nice organization of the tables, which are very accessible and understandable.

The schematic model in Figure 1C is very helpful. Perhaps you could increase the Aa and Ab and put those symbols next to the words “ordered” and “disordered?” Just a suggestion. It would tighten up the model and make it a little easier to connect it to the bar graph. I think that it might be worth putting that schematic in Figure 2 as well, or at least state in the Figure 2 legend that the patterns are shown in the schematic in Figure 1.

Please add the concentration of dynasore used to Table 3.

For Figure 7, could you please clarify in the legend with an explanation of the side-by-side panel pairs in Figure 7B? It is not abundantly clear whether the left panels of each pair are uninfected, or if the panels are overlays – the reader needs to look very closely. The schematic for the treatment protocol is very nice and very helpful.

Author Response

Dear Reviewer,

Thank you very much for the critical reading of the manuscript and the constructive suggestions. We hope that we have addressed properly all your questions and remarks.

In the paper titled “Dynamin inhibitors prevent the establishment of the cytomegalovirus assembly compartment in the early phase of infection” by Štimac et al., the authors do a detailed, temporal analysis of the role that the host protein dynamin may play in the early phases of MCMV infection. The study approach was primarily cell culture infection with well-established inhibitors of dynamin using fluorescence microscopy and protein analysis to chart the formation of the viral assembly complex. While this study is not the first to explore the role of dynamin in CMV infection, the authors do extend the work by exploring the intracellular dynamics at a higher resolution and in finer detail. The exploration of multiple aspects of the Golgi structures is a particularly nice addition to the literature.

Overall, I have to commend the authors for putting together a thoughtful, thorough, interesting manuscript. The storyline is well ordered and explained in a clear way; the figures and tables are extremely well composed; and the findings are important for investigators of CMV as well as those who work on intracellular trafficking. Below are some specific comments on the manuscript.

Specific comments:

I am glad that the researchers included multiple inhibitors of dynamin in the analysis. This is a strength. Even though all of these inhibitors have been well-established in the literature, the fact is that all inhibitors have unique dynamics and the risk of off-effects, and so the bolstering of dynamin inhibition was a good move.

I’m glad that the authors included dynasore-treated uninfected cells in the supplementary materials. However, Figure S1 is a little confusing. There is labeling of “untreated” in the images, but then treated cells at 2, 10, and 40 hours. Were the “untreated” cells really 0-hour treatment? In other words, did you add the virus and the inhibitor and then analyze at a zero timepoint? If so, then “untreated” should be changed to 0-hours, or the table needs to be updated. The left column of the table refers to time, but the images refer to treatment and time. I hope that made sense. Also regarding Figure S1, the panel says 44 hours and the table says 40 hours. A small disconnect, I know – but accuracy will improve the figure.

Thank you for your comment. Figure S1 represents the changes in Golgi patterns in uninfected cells. The purpose was for the inhibitor to act for the same period as in infected cells (this Figure S1 can be compared with Figure 2). Here (Fig S1), the term “Untreated” indeed represents cells that have not been infected nor treated with dynasore. Furthermore, considering that in infected cells (Figure 2), the inhibitor has been added 4 hpi, 2 hrs treatment with dynasore should be comparable to infected cells at 6 hpi, 10 hpi of treatment should be comparable to infected cells at 14 hpi, and 44 hrs should be comparable to infected cells at 48 hpi. Generally, in immunofluorescence studies, we used untreated cells as an indicator of the 0-hour treatment. Namely, we have performed pre-testing, which shown no difference whether cells were infected (with or without inhibitor) and then analyzed at zero points, or uninfected. However, in Western blot experiments, 0 hpi is indicated because in that case, cells were infected (again with or without inhibitor) and then analyzed at zero points. We also tested uninfected cells for WB, but there was no difference (not shown).

We have corrected the table, and 44 hrs are valid. We apologize for that mistake. Of course, it is important for the proper kinetics to be noticed. The left column of the table has also been corrected, the term “dynasore” has been added, and the table now represents both treatment and time. Finally, in line with your comment, we have restricted reporting to 1 significant digit for all our immunofluorescence measurements. That was also our opinion while writing the manuscript.

Note that in Figure S2, the title of the figure suggests that the cells were pre-treated with dynasore, but they were actually treated alongside infection. Rather than saying that they were treated before infection, I would say that they were treated before viral uptake or viral internalization. I think that might be more accurate. I believe that the authors may have been using the term “infection” and “uptake” synonymously? Please confirm.

Thank you for the correction. The title of the figure is corrected to: “Dynasore inhibits the establishment of the preAC when added before viral internalization.”

I’m glad that the authors did multiple analyses of the EE-to-ERC stage with multiple markers (Rab11, Arf6, Rab10). These were key experiments for the project.

Thank you for the comment.

The exposures for some of the microscopy seems a little high. I realize that this is not something that can be remedied in retrospect, and I don’t think that it is a problem for analysis, since the authors did a nice job of quantifying their results. It is certainly just an inherent problem with reporting microscopy.

You are correct that in the pericentriolar area, the exposures can be detected as overexposed. However, there are two reasons for that: (1) We have intended to keep the same parameters as in uninfected cells (where the signal was lower, or even zero (as for Rab10, Arf6, Vps34). Considering such a high over-recruitment of the majority of markers in the pericentriolar area in infected cells, the signal was so high. However, in these cases, when we need to explore the structural changes, we take the picture at the same parameters as in control cells, but also, in parallel, with lower parameters. (2) in MCMV infected cells, the intensity of fluorescence is quite different between the pericentriolar area and cell cytoplasm, especially subplasmalemmal. Therefore, when we decrease parameters to get ideal signal in the pericentriolar area, we usually lose the cytosolic and subplasmalemmal signal.

The writing overall is very good, although there are some areas where the grammar is a bit awkward. The manuscript could use one last round of editing for grammar. Also, there is a mix of using commas versus decimal points in some of the numeric reporting, and that should definitely be cleaned up before publication. However, I would note that the “storytelling” in the Results is very good—I appreciate that the authors explained the significance of each of the markers they assessed along with the significance of the markers.

Thank you for that remark. Considering that English is not our native language, we have checked the whole manuscript using Grammarly, but unfortunately some mistakes escaped detection. We also corrected the commas with decimal points. In Croatian writing, we use commas, and here we have adapted to using decimal points.

Regarding the cell compartment markers, I reviewed the main markers that the authors chose, and it seems that the authors did the best job of choosing well-established markers and antibodies as is possible. So I can’t really see any problems in terms of quality control. I am going on the good faith reporting of the researchers—they seem to have covered all the bases, as far as I can tell.

Thank you for that comment. In our previous work (Lučin et al, Front Cell Dev Biol, 2020), we explored the expression of 64 markers in the early phase of MCMV infection, which was a good basis for this manuscript.

I don’t want to suggest unnecessary extra work for the authors, because the manuscript is so nice. But a summary schematic might be nice? Just a suggestion—not a requirement. I mention it because the authors included so many nice schematics in the figures.

We also share an opinion that a schematic would be nice, but we have a couple of questions to address to fill it appropriately. Also, we plan to write a review where schematic figures will find their places.

A few minor comments:

The reporting of percentages out to 2 significant figures is perhaps a bit beyond the accuracy of the microscopy measurements. It’s hard to imagine 0.01 cellular at this resolution. I think that reporting out to 1 significant digit for these measurements might be sufficient – but it’s not crucial, and I know that this is often the norm in other publications. I will defer to the editor in regard to the reporting preferences of the journal for fluorescence microscopy.

We agree with you. As explained before, we have changed all values in tables and text to only 1 significant digit for immunofluorescence results.

The term “data” is normally plural in the scientific literature. Again, I defer to the journal’s house style.

The phrase “over-recruitment” might be easier to read if it is hyphenated. While it is not acting as an adjective, it is a bit of a coined term, so the punctuation might make it easier on the reader.

Thank you for the suggestion. The phrase “over-recruitment” has been hyphenated in the manuscript.

Some minor comments on Figures & Tables:

The figures and tables are laid out and sized very nicely. I am particularly grateful for the very nice organization of the tables, which are very accessible and understandable.

Thank you for that comment. We were aware that there are lot of data, and we wanted to present them in the best possible way.

The schematic model in Figure 1C is very helpful. Perhaps you could increase the Aa and Ab and put those symbols next to the words “ordered” and “disordered?” Just a suggestion. It would tighten up the model and make it a little easier to connect it to the bar graph. I think that it might be worth putting that schematic in Figure 2 as well, or at least state in the Figure 2 legend that the patterns are shown in the schematic in Figure 1.

Thank you for the suggestion. The Aa and Ab were adjusted according to your instructions in Figure 1. The schematic has also been incorporated into Figures 2 and S2. The symbols are now present both in the schema and in the graph.

Please add the concentration of dynasore used to Table 3.

Thank you for the correction. The concentration of the dynasore has been added.

For Figure 7, could you please clarify in the legend with an explanation of the side-by-side panel pairs in Figure 7B? It is not abundantly clear whether the left panels of each pair are uninfected, or if the panels are overlays – the reader needs to look very closely. The schematic for the treatment protocol is very nice and very helpful.

Thank you for the comment. We have improved Figure 7 and hope that it is now better understood.

Reviewer 2 Report

This study describes the effect of Dynamin inhibition during MCMV infection, specifically during the early phase of infection when an assembly compartment (AC) is generated.

The scope and impact of this work are somewhat limited due to the fact that the effect of dynamin inhibition has been extensively studied during other phases of CMV infection, such as endocytic entry and egress.

Within this narrow scope, however, this study was done with great scientific rigor and well-described research questions and objectives, as described in lines 67-71.

Technical comments:

The Methods section 2.3 has a detailed description of how image analysis was done. Can the authors expand on their exact method of quantification of the % of early endosomes present in the AC? Currently, Lines 190-191 describe:” any fluorescent signal that was concentrated in the pericentriolar area”. In general, the representative images throughout the different figures are visually in accordance with the amounts quantified in bar diagrams below. However, for the staining with EEA1 (Figure 4A), this is less obvious. A simple expanded clarification of how the quantification in Figure 4 was performed will suffice.

The authors observe in Figure 2 that dynasore treatment results in a shift in golgi organization pattern from A (ordered) to Ab (disordered), but blocks progression to B- or C- patterns.

In follow-up experiments, the authors describe that this also occurs in mock-treated cells. Earlier addition of dynasore, did not affect the kinetics of infection and AC formation (Supplementary Figure 2), providing a decent argument that this dynasore-induced fragmentation did not change the observations pertaining to the role of dynamin in CMV infection. However, could the authors expand on how this effect of dynamin inhibitors on uninfected cells might affect their future application for treatment of cancer?

The effect on AC development was consistent with the use of multiple dynamin inhibitors. Is the same true for the effect on virion and protein production (Figure 9)? In addition, the authors describe Dyngo-4a as :” an improved dynasore analog characterized by less toxicity and less unspecific reactions” (lines 379-381). Is there any reason that this inhibitor was not used throughout the study instead of dynasore?

Figure 8: two replicate experiments were done, could the authors add error bars to the bar diagrams?

Minor textual revisions:

Line 36:  “after congenital infections and immunocompromised patients”

Suggested edit: “after congenital infections and infections in immunocompromised patients”.

Line 37: - “The fatal” – remove “the”

 - “The vaccine”: substitute with “A vaccine”

38: “The understanding”: either “the understanding of”, or “understanding the”

56: “viral structural proteins expression”: replace with: “expression of viral structural proteins”.

259: “recruit”, replace with “recruits”.

474: The term “over recruitment” is used multiple times throughout the manuscript. I would suggest substituting with “over-recruitment” to improve readability.

685: “reorganization”, replace with “reorganizations”.

727: ”at membrane of infected cells”, replace with: “at the membrane of infected cells”.

775: “ Even more, the dynamin over activation”, replace with “ Even more, dynamin over-activation”

820: “are our observation”, change to either “is our observation” or are our observations”

Author Response

Dear Reviewer,

Thank you very much for the critical reading of the manuscript and the constructive suggestions. We hope that we have answered all your questions and remarks. Furthermore, we have again checked the manuscript by Grammarly to improve English.

This study describes the effect of Dynamin inhibition during MCMV infection, specifically during the early phase of infection when an assembly compartment (AC) is generated.

The scope and impact of this work are somewhat limited due to the fact that the effect of dynamin inhibition has been extensively studied during other phases of CMV infection, such as endocytic entry and egress.

Within this narrow scope, however, this study was done with great scientific rigor and well-described research questions and objectives, as described in lines 67-71.

Technical comments:

The Methods section 2.3 has a detailed description of how image analysis was done. Can the authors expand on their exact method of quantification of the % of early endosomes present in the AC? Currently, Lines 190-191 describe:” any fluorescent signal that was concentrated in the pericentriolar area”. In general, the representative images throughout the different figures are visually in accordance with the amounts quantified in bar diagrams below. However, for the staining with EEA1 (Figure 4A), this is less obvious. A simple expanded clarification of how the quantification in Figure 4 was performed will suffice.

Thank you for the comment. We have also observed that EEA1 shows less characteristic accumulation than other EE markers. The accumulation increases at later time points of infection, especially in the late phase of infection (48 hpi). Importantly, our unpublished observations show a decrease of EEA1 expression (Western blot) and recovery in the later phases of infection. A similar pattern was observed in HCMV infected cells. However, to clarify the results, we have integrated schematic presentation of the patterns considered as o-preAC: the point is α £ 90°, i.e., the angle that closes the accumulated vesicles should be at least 90°. The explanations are also added to the text: Materials and Methods (chapter 2.3, lines 190-191 (191-192 in pdf with Track changes)) and Results (chapter 3.4, lines 473-474 and 481-483 (716-717 and 725-727 in pdf with Track changes)).

The authors observe in Figure 2 that dynasore treatment results in a shift in golgi organization pattern from A (ordered) to Ab (disordered), but blocks progression to B- or C- patterns.

In follow-up experiments, the authors describe that this also occurs in mock-treated cells. Earlier addition of dynasore, did not affect the kinetics of infection and AC formation (Supplementary Figure 2), providing a decent argument that this dynasore-induced fragmentation did not change the observations pertaining to the role of dynamin in CMV infection. However, could the authors expand on how this effect of dynamin inhibitors on uninfected cells might affect their future application for treatment of cancer?

That is an interesting question. We have observed two important points following treatment of uninfected cells with dynasore: (1) increase in the ratio of cells with disordered phenotype, and (2) Golgi vacuolization (Figure S2). We have not investigated these effects further, but some speculations can be derived. Considering that Golgi unlinking precedes cell division, increasing cell ratio with disordered Golgi can indicate some disturbance in cell division. Furthermore, Golgi fragmentation can be a consequence of dynamin depletion and after disturbance of microtubule dynamics (Tanabe and Takei, J Cell Biol, 2009). As is well known, normal microtubule dynamics are also important for normal regulation of the cell cycle. Otherwise, Golgi vacuolization is most probably the consequence of inhibition of detaching of vesicles from the Golgi. Some of them are important for delivering cell glycoproteins to the cell surface. Therefore, disturbing Golgi may impair the delivery of some essential surface molecules and seriously disturb cellular homeostasis that could end with cell death. So, both effects can be interesting for cancer therapists. We have not discussed this area because it is out of the scope of this manuscript.

The effect on AC development was consistent with the use of multiple dynamin inhibitors. Is the same true for the effect on virion and protein production (Figure 9)? In addition, the authors describe Dyngo-4a as :” an improved dynasore analog characterized by less toxicity and less unspecific reactions” (lines 379-381). Is there any reason that this inhibitor was not used throughout the study instead of dynasore?

We have not tested other inhibitors on virion and protein production, but we will test the most important experiments with Dyngo-4a, Dynole-34-2, and MitMAB because we will continue with the investigation of dynamin and those inhibitors.

Dyngo-4a is described in the literature as “an improved dynasore analog characterized by less toxicity and less unspecific reactions,” but it is also more expensive than dynasore. Considering that the effect of dynasore was powerful, sometimes even total (especially 6 hpi), and that the viability was similar to Dyngo-4a (Table 2), the advantage to Dyngo-4a did not seem so significant for this phase of experiments.

Figure 8: two replicate experiments were done, could the authors add error bars to the bar diagrams?

Thank you for the suggestion. The error bars have been added in Figure 8, and significance to the untreated sample has been found for M57 expression 24 hpi and 48 hpi (student’s t-test).

Minor textual revisions:

Line 36:  “after congenital infections and immunocompromised patients”

Suggested edit: “after congenital infections and infections in immunocompromised patients”.

Thank you for the suggestion. The text was corrected.

Line 37: - “The fatal” – remove “the”

 - “The vaccine”: substitute with “A vaccine”

 Thank you for the suggestion. The text was corrected.

38: “The understanding”: either “the understanding of”, or “understanding the”

 Thank you for the suggestion. The text was corrected.

56: “viral structural proteins expression”: replace with: “expression of viral structural proteins”.

 Thank you for the suggestion. The text was corrected.

259: “recruit”, replace with “recruits”.

Thank you for the suggestion. The text was corrected.

474: The term “over recruitment” is used multiple times throughout the manuscript. I would suggest substituting with “over-recruitment” to improve readability.

Thank you for the suggestion. The term was corrected.